# Cellular assays identify barriers impeding iron-sulfur enzyme activity in a non-native prokaryotic host

Francesca D'Angelo[1†§], Elena Fernández-Fueyo[2†§], Pierre Simon Garcia[1,3†§], Helena Shomar[2†§], Martin Pelosse[4], Rita Rebelo Manuel[2], Ferhat Büke[2], Siyi Liu[5], Niels van den Broek[2], Nicolas Duraffourg[4], Carol de Ram[6], Martin Pabst[6], Emmanuelle Bouveret[1], Simonetta Gribaldo[3], Béatrice Py[5], Sandrine Ollagnier de Choudens[4], Frédéric Barras[1*‡], Gregory Bokinsky[2*‡]

[1]Unit Stress Adaptation and Metabolism of Enterobacteria, Department of Microbiology, Université de Paris, UMR CNRS 2001, Institut Pasteur, Paris, France; [2]Department of Bionanoscience, Kavli Institute of Nanoscience, Delft University of Technology, Delft, Netherlands; [3]Institut Pasteur, Université de Paris, CNRS UMR6047, Evolutionary Biology of the Microbial Cell, Department of Microbiology, Paris, France; [4]Univ. Grenoble Alpes, CNRS, CEA, IRIG, Laboratoire de Chimie et Biologie des Métaux, Grenoble, France; [5]Aix-Marseille Université-CNRS, Laboratoire de Chimie Bactérienne UMR 7283, Institut de Microbiologie de la Méditerranée, Institut Microbiologie Bioénergies Biotechnologie, Marseille, France; [6]Department of Biotechnology, Delft University of Technology, Delft, Netherlands

*For correspondence:
frederic.barras@pasteur.fr (FB);
g.e.bokinsky@tudelft.nl (GB)

[†]These authors contributed equally to this work
[‡]These authors also contributed equally to this work

[§]These authors are listed alphabetically

Competing interest: The authors declare that no competing interests exist.

**Abstract** Iron-sulfur (Fe-S) clusters are ancient and ubiquitous protein cofactors and play irreplaceable roles in many metabolic and regulatory processes. Fe-S clusters are built and distributed to Fe-S enzymes by dedicated protein networks. The core components of these networks are widely conserved and highly versatile. However, Fe-S proteins and enzymes are often inactive outside their native host species. We sought to systematically investigate the compatibility of Fe-S networks with non-native Fe-S enzymes. By using collections of Fe-S enzyme orthologs representative of the entire range of prokaryotic diversity, we uncovered a striking correlation between phylogenetic distance and probability of functional expression. Moreover, coexpression of a heterologous Fe-S biogenesis pathway increases the phylogenetic range of orthologs that can be supported by the foreign host. We also find that Fe-S enzymes that require specific electron carrier proteins are rarely functionally expressed unless their taxon-specific reducing partners are identified and co-expressed. We demonstrate how these principles can be applied to improve the activity of a radical S-adenosyl methionine(rSAM) enzyme from a *Streptomyces* antibiotic biosynthesis pathway in *Escherichia coli*. Our results clarify how oxygen sensitivity and incompatibilities with foreign Fe-S and electron transfer networks each impede heterologous activity. In particular, identifying compatible electron transfer proteins and heterologous Fe-S biogenesis pathways may prove essential for engineering functional Fe-S enzyme-dependent pathways.

## Introduction

Many enzymes depend upon protein networks that perform essential tasks such as cofactor distribution, membrane insertion, or covalent modification. These cellular networks are evolutionarily ancient and have diversified considerably through adaptation and coevolution with their client enzymes. As a consequence, cellular networks may be unable to recognize client enzymes from foreign species.

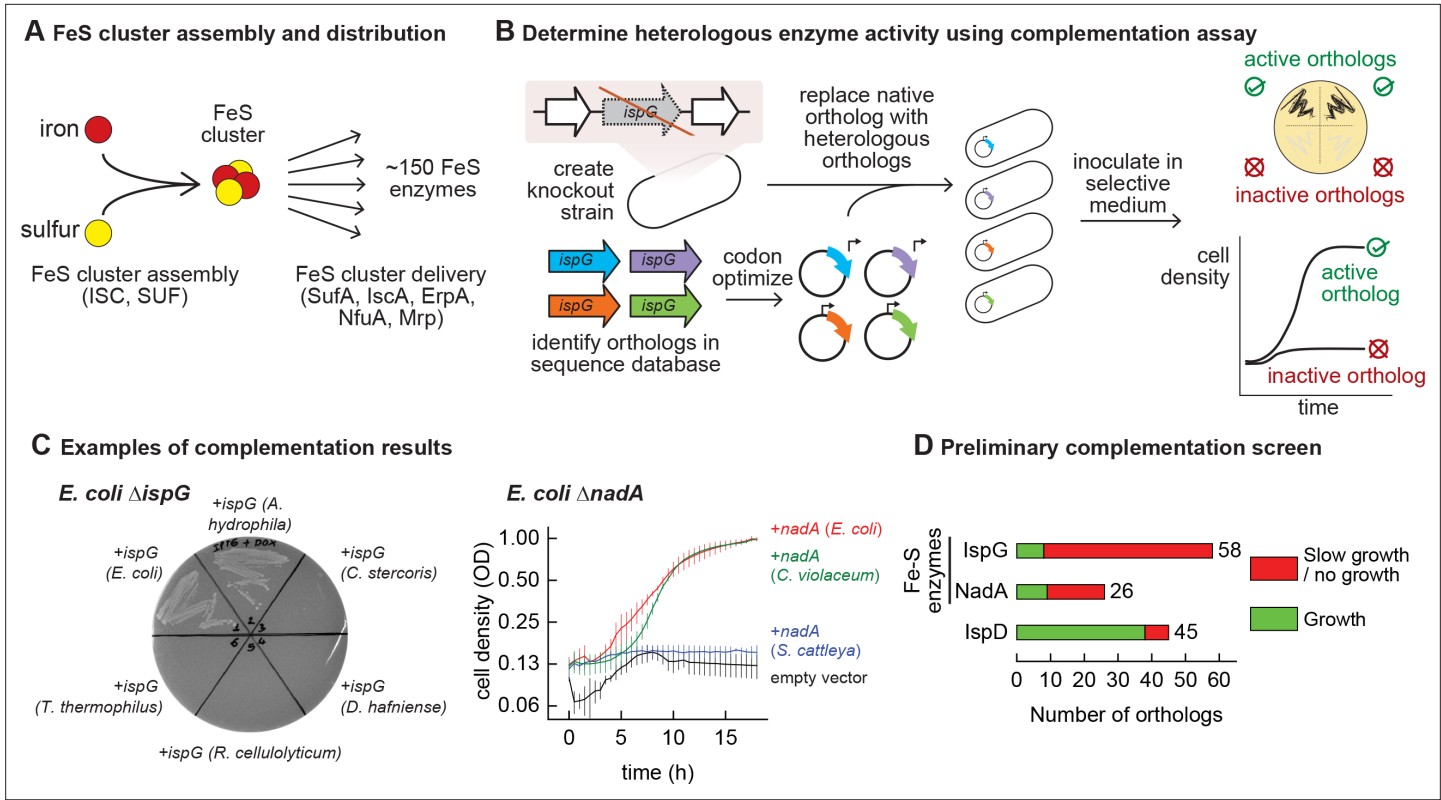

**Figure 1.** Mapping functional expression of heterologous iron-sulfur (Fe-S) enzymes using complementation experiments. (**A**) In prokaryotes, two pathways that assemble and deliver Fe-S clusters have been identified (ISC, SUF), as have several Fe-S delivery proteins (SufA, IscA, ErpA, NfuA in *Escherichia coli*). (**B**) The complementation assays used to determine activity of heterologous orthologs expressed by *E. coli*. Orthologs of conditionally essential enzymes are identified in sequence databases, codon optimized, and cloned into a low-copy plasmid. Expression is controlled by an inducible promoter ($P_{Tet}$). The plasmids are then transformed into *E. coli* strains lacking the corresponding *E. coli* ortholog. Each strain is inoculated in selective media and expression of the heterologous ortholog is induced. Growth in selective media indicates that the heterologous ortholog retains sufficient activity to support growth of *E. coli*. (**C**) Exemplary results of complementation assay obtained using either solid or liquid selective medium. Error bars represent standard error of duplicate cultures inoculated from separate colonies. (**D**) Results of preliminary complementation tests comparing activity of Fe-S enzymes with a non-Fe-S enzyme. Orthologs tested for the preliminary screen are listed in **Supplementary file 1**.

Thus, many network-dependent enzymes often show limited or no activity when expressed within foreign species, and genes encoding such enzymes may be less likely to persist following horizontal transfer (*Porse et al., 2018*). Because the biosynthesis of many valuable natural products relies upon network-dependent enzymes, their compatibility with non-native cellular networks is an important question for bioengineering.

Iron-sulfur (Fe-S) enzymes depend upon cellular Fe-S biogenesis networks to acquire Fe-S cluster cofactors (*Johnson et al., 2005*). These networks assemble and distribute Fe-S clusters via Fe-S scaffold and carrier proteins (*Figure 1A*). Two Fe-S biogenesis pathways (ISC and SUF) have been identified in prokaryotes. Common to both ISC and SUF is the use of cysteine desulfurase enzymes (IscS or SufSE) to obtain sulfur from L-cysteine, and scaffold proteins (IscU or SufBCD) to assemble Fe-S clusters. Fe-S cluster carrier proteins of the A-type (IscA, SufA, ErpA), Nfu-type (NfuA), or ApbC-type (Mrp) deliver clusters to apo targets (*Py and Barras, 2010*; *Roche et al., 2013*). Fe-S biogenesis systems exemplify the diversification of essential cellular networks (*Roche et al., 2013*). It has been proposed that the diversification of Fe-S enzymes and networks is driven by adaptations to molecular oxygen, which reduces the environmental availability of iron and is destructive to Fe-S clusters (*Andreini et al., 2017*; *Boyd et al., 2014*; *Imlay, 2006*). Comparisons between SUF pathways, which appear to be more widespread, reveal further diversification: while core SUF components (SufB and SufC) are conserved, SUF systems differ in the number of proteins used to assist Fe-S cluster assembly, delivery, and maturation (e.g. SufD, SufE, SufS, SufU, SufT) (*Boyd et al., 2014*; *Garcia et al., 2019*).

Fe-S enzymes often show no activity when expressed within foreign hosts (*Lanz et al., 2012*; *Nakai et al., 2015*; *Nakamura et al., 1999*). As with any heterologous enzyme, inactivity in vivo may arise from poor expression, i.e., inefficient transcription or translation. In a bioengineering context, these barriers are routinely overcome using tools such as strong promoters and codon optimization. Unfortunately, even if heterologous enzymes are expressed, few tools exist for ensuring activity is retained. Particularly for Fe-S enzymes, incompatibilities with foreign Fe-S biogenesis networks that arise from diversification may also limit activity. Here, we combine bioinformatics and phylogeny with an in vivo platform to systematically survey the ability of two types of Fe-S enzymes to retain activity within a foreign host. In doing so, we assessed how environmental adaptations and the dependencies of Fe-S enzymes upon cellular networks each contribute to limiting heterologous activity in prokaryotic hosts. We show that the activity of heterologous Fe-S enzymes can be recovered by broadening compatibility of host networks and by modifying growth conditions. This survey leads us to further investigate these barriers and enhance in vivo activity of a cobalamin-dependent radical *S*-adenosyl methionine (rSAM) methyltransferase enzyme involved in antibiotic biosynthesis.

## Results

### Heterologous Fe-S enzymes are less likely to be functionally expressed than non-Fe-S enzymes

We performed a preliminary screen to evaluate the ability of Fe-S enzymes to retain activity within a heterologous host (*Figure 1B*). We constructed two knockout strains of *E. coli* MG1655 that each lack a conditionally essential Fe-S enzyme: NadA (quinolinate synthase) and IspG (4-hydroxy-3-methylbut-2-enyl-diphosphate synthase). These proteins were chosen because they are involved in metabolic pathways for valuable compounds (e.g. vitamins, biofuels, and fragrances). Orthologs of these enzymes were identified in genomes belonging to a diverse range of bacterial phyla (e.g. *Actinobacteria*, *Firmicutes*, *Proteobacteria*, *Cyanobacteria*) for a preliminary survey (*Supplementary file 1*). The activities of the heterologous enzymes were tested using a complementation assay within the corresponding *E. coli* knockout strain: growth in selective media indicates that the heterologous enzyme is functional when expressed in *E. coli* (*Figure 1C*). To avoid complementation failures arising from poor enzyme expression (due to e.g. weak promoters or inefficient translation), all heterologous genes were codon optimized, cloned into a multicopy expression vector, and transcribed from a chemically inducible promoter. The majority of Fe-S enzyme orthologs (65 out of 84) failed to complement growth (*Figure 1D*, *Supplementary file 1*). For comparison, we tested orthologs of a non-Fe-S enzyme (IspD, 2-C-methyl-D-erythritol 4-phosphate cytidylyltransferase, which belongs to the same biosynthetic pathway as IspG) for activity in *E. coli*. The majority of IspD orthologs (38 out of 45) complemented growth of *E. coli* Δ*ispD*, including many orthologs from species whose IspG orthologs failed to complement growth of *E. coli* Δ*ispG* (*Supplementary file 1*). Of note, average sequence identity of the heterologous IspG and IspD orthologs with the *E. coli* orthologs was 45% and 36%, respectively. These observations suggest that Fe-S enzymes are far more likely than non-Fe-S enzymes to lose activity within a heterologous host.

### Correlating compatibility of heterologous enzymes with phylogenetic distance from host ortholog

We next sought to explore the factors that determine whether an Fe-S enzyme becomes inactive within a heterologous host. We first built a protein database that is representative of known prokaryotic diversity, comprising 248 prokaryotic (archaeal and bacterial) proteomes corresponding to 48 phyla, 96 classes, and 218 orders (*Supplementary file 2*). This database contains 65 proteomes that were used for preliminary screen. We identified 192 and 196 orthologs of NadA and IspG, respectively (*Supplementary file 3*), which were used to infer a maximum likelihood phylogeny (*Figure 2A and B*). Next, we used a clustering approach based on phylogenetic distances to select representative sequences in a phylogenetically based dataset to be experimentally tested, which yielded 47 sequences for NadA and IspG each (*Supplementary files 3 and 4*). The distribution of patristic distances of the selected orthologs from the *E. coli* NadA and IspG sequences closely matched with the patristic distance distribution obtained from all enzyme homologs, indicating that the selected orthologs are representative of the entire database of known orthologs (*Figure 2C*). Furthermore, a

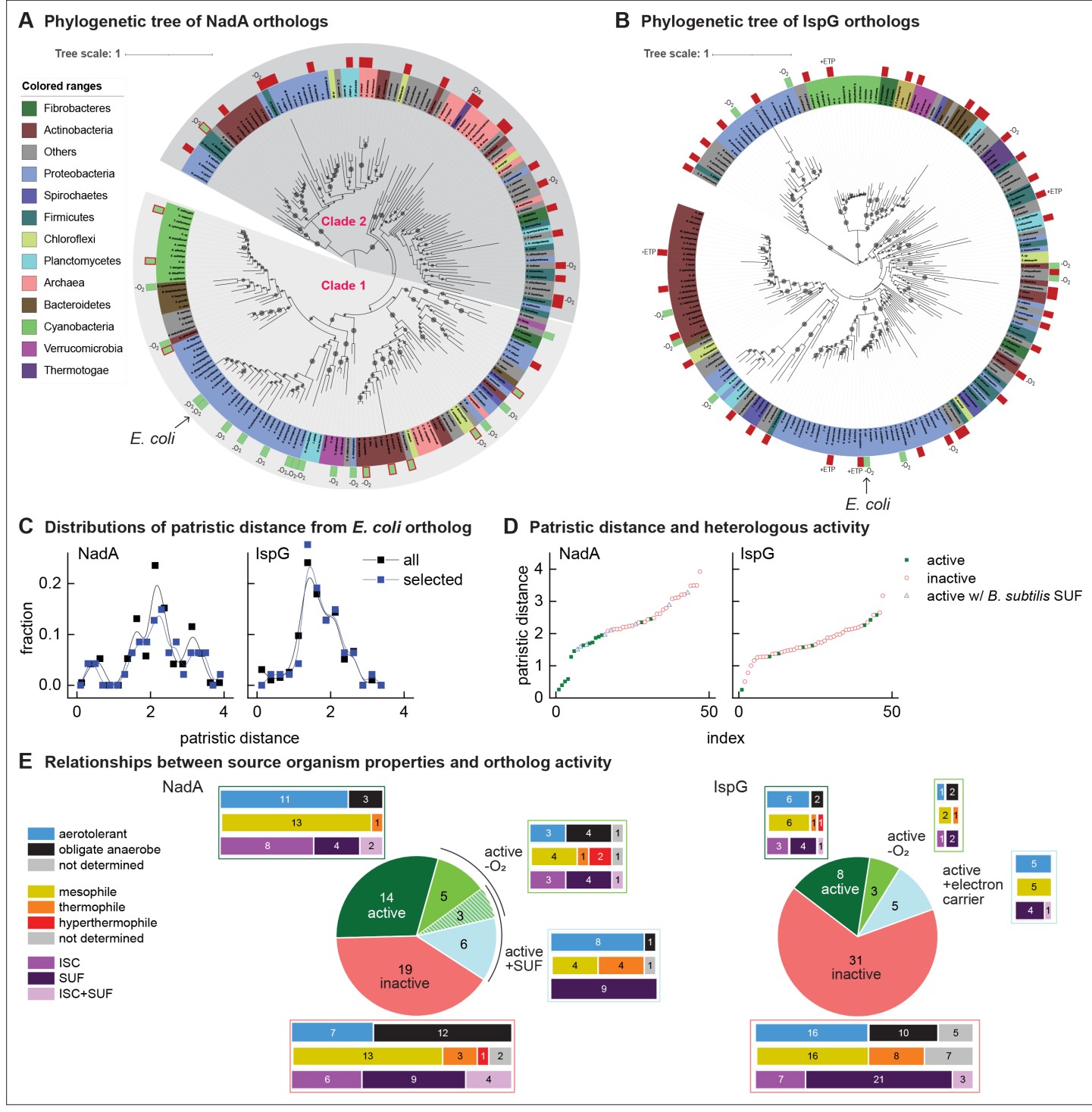

**Figure 2.** Phylogenetic diagrams contrasting the compatibility of iron-sulfur (Fe-S) enzymes NadA and IspG across a selection of orthologs representative of known prokaryotic sequence diversity. (**A, B**) Phylogeny of NadA orthologs (**A**) and IspG orthologs (**B**). In each diagram, the position of *Escherichia coli* ortholog is highlighted by an arrow. Orthologs selected for our complementation assay are indicated by color rectangles (green, successful complementation; green surrounded by red, complementation by coexpression of *Bacillus subtilis* SUF; red, no complementation). Orthologs complementing growth in anaerobic conditions are indicated by –O₂. IspG orthologs recovered by expression of compatible electron transfer protein indicated by +ETP. NadA Clades 1 and 2 are indicated. (**C**) Distributions of patristic distances from the corresponding *E. coli* ortholog of all sequences (black curves) and the sequences of orthologs selected for testing (blue curves). (**D**) Complementation and reactivation results correlated with patristic distance from the *E. coli* ortholog. (**E**) Summary of complementation and recovery results for heterologous orthologs. The bar plots adjoining each

*Figure 2 continued on next page*

*Figure 2 continued*

wedge indicate the characteristics of native hosts (aerotolerance, optimal growth temperature, Fe-S cluster biosynthesis system). SUF and ISC indicate that homologs to SufBD or IscU, respectively, are identified in native genome. Results and properties of individual orthologs are given in *Tables 1 and 2*.

The online version of this article includes the following figure supplement(s) for figure 2:

**Figure supplement 1.** Growth curves of *Escherichia coli ΔnadA* grown in M9, Kan, Amp, aTc, and isopropyl β-D-1-thiogalactopyranoside complemented with pBbS2k and pBbA5a plasmids carrying different genes: empty vectors (blue line), pBbS2k and *Bacillus subtilis* SUF (red line), *B. subtilis nadA* and pBbA5a (green line), and *B. subtilis nadA* and *B. subtilis* SUF (black line).

**Figure supplement 2.** *Escherichia coli* ErpA mediates iron-sulfur (Fe-S) cluster transfer into heterologous IspGs.

**Figure supplement 3.** Confirming expression of heterologous orthologs.

**Figure supplement 4.** Biochemical detection of iron-sulfur (Fe-S) clusters within purified NadA and IspG proteins.

two-sided Wilcoxon test between full and sampled distances from *E. coli* rejected the hypothesis of significant differences between the selected set of orthologs (47 both) and the set of all orthologs (191 for NadA and 195 for IspG, p=0.4714 for NadA and p=0.4739 for IspG). Altogether, these results indicate that the phylogenetically based set is a representative sample of the phylogenetic diversity of NadA and IspG sequences across prokaryotes.

All orthologs from the phylogenetically based sets of IspG and NadA were each tested using the complementation assay in aerobic conditions at 37°C. Among the NadA orthologs representative of phylogenetic diversity, 14 out of 47 orthologs complemented growth of *E. coli ΔnadA*, indicating functional expression (*Table 1*). Consistent with our hypothesis, NadA orthologs that were functionally expressed in *E. coli* exhibited a low patristic distance from the *E. coli* NadA ortholog (*Figure 2D*). Repeating the complementation assay at a lower temperature (28°C) did not affect the outcome. The high correlation ratio ($\eta$) between heterologous expression and patristic distance ($\eta = 0.61$) and two-sided Wilcoxon test (p<0.01) suggest that phylogenetic proximity to the *E. coli* ortholog is a useful predictor of whether a NadA ortholog can be functionally expressed in *E. coli*.

Among the 47 IspG orthologs selected from the phylogenetically based set, only 8 complemented growth of *E. coli ΔispG* (*Table 2*). The sequences of functionally expressed IspG orthologs did not group together in the IspG phylogeny but were instead patchily dispersed across the entire phylogenetic distribution of sequences (*Figure 2D*). Accordingly, no correlation between functional expression and patristic distance from *E. coli* was observed ($\eta = 0.03$, two-sided Wilcoxon test p>0.01). Altogether, these results indicate that the activity of heterologous IspG orthologs in *E. coli* does not correlate with phylogenetic proximity, in contrast with NadA orthologs.

## Aerobic conditions impede activity of heterologous Fe-S enzymes

As Fe-S clusters can be altered by reactive oxygen species, the aerobic conditions used in our complementation assays may inactivate some orthologs, particularly those obtained from obligate anaerobes. We therefore repeated our complementation experiments in anaerobic conditions. Anaerobic growth enabled eight additional NadA orthologs to complement growth of *E. coli ΔnadA*. Five of the 8 NadA orthologs recovered in anaerobic conditions were obtained from obligate anaerobes (*Table 1*, *Figure 2A and E*). However, anaerobiosis recovered only three additional IspG orthologs (*Table 2*, *Figure 2B and E*). All NadA and IspG orthologs that complemented growth in aerobic conditions also complemented growth in anaerobic conditions. Surprisingly, the aerotolerance of the hosts from which the orthologs were obtained is not an absolute predictor of activity in aerobic or anaerobic conditions (*Tables 1 and 2*, *Figure 2E*).

## Recovery of NadA orthologs using a heterologous Fe-S biogenesis pathway

We next tested whether coexpression of a heterologous Fe-S pathway might reactivate heterologous Fe-S enzymes. We cloned the *Bacillus subtilis* SUF operon into a separate vector (pBbA5a), which was introduced into *E. coli ΔnadA* expressing *B. subtilis* NadA (*Supplementary file 5*). Coexpression of the *B. subtilis* SUF system together with *B. subtilis* NadA fully recovered growth of *E. coli ΔnadA* in aerobic conditions (*Figure 2—figure supplement 1*, *Table 1*). Expression of *B. subtilis* SUF activated eight additional NadA orthologs. Interestingly, three orthologs recovered by *B. subtilis* SUF were also recovered in anaerobic conditions (*Table 1*, *Figure 2A and E*). By comparison, overexpression

**Table 1.** Compiled results of complementation and protein expression experiments for each NadA ortholog. Complementation of *Escherichia coli ΔnadA* and recovery of NadA activity with either SUF coexpression (pNadA + pBsSUF for *Bacillus subtilis* SUF, or pNadA + pEcSUF for *E. coli* SUF) or anaerobic growth (pNadA −O2). Minus (−), negative complementation; plus (+), positive complementation; N.T., not tested. Protein expression, predicted translation initiation rate from the RBS calculator (log10 scale) and outcome of SDS-PAGE and MS detection experiments. SDS-PAGE column, (+) band observed on an SDS-PAGE gel. MS column, (+) confirmed detection by mass spectroscopy; N.T., not tested; detection N.T., negative complementation and detection by MS not tested. The properties of native species for each ortholog are indicated: O2 tolerance: (+) aerotolerant, (−) obligate anaerobe, (?) unknown; iron-sulfur (Fe-S) synthesis system describes homologs to IscU or SufB identified in native genome. SufBD, SufB homolog detected; IscU, IscU homolog detected; SufBC, IscU, both systems detected.

| Phylum | Species/strain | Complementation/recovery | | | | | Protein expression | | | | Native host characteristics | | Fe-S synthesis system |
|---|---|---|---|---|---|---|---|---|---|---|---|---|---|
| | | pNadA | pNadA + pBsSUF | pNadA + pEcSUF | pNadA −O2 | Complemented or recovered | Predicted translation initiation (log10) | SDS-PAGE | MS | Complemented/ recovered /detected | O2 tolerance | Temperature | |
| Actinobacteria | Mycolicibacterium smegmatis MC2 155 | − | + | + | + | + | 3.3 | − | + | + | + | Mesophilic | SufBD |
| | Rubrobacter xylanophilus DSM 9941 | + | N.T. | N.T. | + | + | 2.7 | − | N.T. | + | + | Thermophilic | SufBD |
| | Collinsella stercoris DSM 13279 | − | − | − | − | − | 3.4 | − | N.T. | Detection N.T. | − | Mesophilic | SufBD |
| | Acidimicrobium ferrooxidans DSM 10331 | − | + | − | − | + | 2.2 | − | N.T. | + | + | Thermophilic | SufBD |
| | Streptomyces cattleya NRRL 8057 = DSM 46488 | − | − | − | − | − | 4.8 | − | + | + | + | Mesophilic | SufBD |
| Bacteroidetes | Saliniriga cyanobacteriivorans | + | N.T. | N.T. | + | + | N.T. | + | N.T. | + | − | Mesophilic | SufBD |
| Candidatus Fermentibacteria | Candidatus Fermentibacter daniensis | − | − | − | − | − | 3.9 | − | N.T. | Detection N.T. | − | Mesophilic | IscU |
| Candidatus Melainabacteria | Candidatus Gastranaerophilales bacterium HUM_7 | − | − | − | + | + | 3.8 | − | N.T. | + | − | ? | IscU |
| Chloroflexi | Sphaerobacter thermophilus DSM 20745 | − | + | − | + | + | 4.1 | − | N.T. | + | + | Thermophilic | SufBD |
| | Thermogemmatispora tikiterensis | − | − | − | − | − | 2.8 | − | N.T. | Detection N.T. | + | Thermophilic | SufBD |
| Chrysiogenetes | Desulfurispirillum indicum S5 | − | − | − | + | + | 3.6 | + | N.T. | + | − | Mesophilic | IscU |
| Cyanobacteria | Synechocystis sp. PCC 6803 | − | + | − | − | + | 3.3 | − | N.T. | + | + | Mesophilic | SufBD |
| | Crocosphaera watsonii WH 8501 | − | + | − | − | + | 2.8 | − | N.T. | + | + | ? | SufBD |
| Deinococcus-Thermus | Thermus thermophilus HB27 | − | + | − | − | + | 3.5 | − | N.T. | + | + | Thermophilic | SufBD |
| Dictyoglomi | Dictyoglomus thermophilum H-6-12 | − | − | − | + | + | 3.6 | + | N.T. | + | − | Hyperthermophilic | SufBD, IscU |
| | Methanopyrus kandleri AV19 | − | − | − | − | − | 2.9 | − | N.T. | Detection N.T. | − | Hyperthermophilic | SufBD |
| | Haloferax volcanii DS2 | − | − | − | − | − | 3.1 | − | N.T. | Detection N.T. | + | Mesophilic | SufBD |
| | Haloterrigena turkmenica DSM 5511 | − | − | − | − | − | 3.8 | − | N.T. | Detection N.T. | + | Mesophilic | SufBD |
| Euryarchaeota | Candidatus Methanomethylophilus alvus Mx1201 | − | − | − | − | − | 3.0 | − | N.T. | Detection N.T. | − | ? | SufBD |
| Fibrobacteres | Chitinivibrio alkaliphilus ACht1 | − | − | − | − | − | 3.2 | − | N.T. | Detection N.T. | − | Mesophilic | IscU |

*Table 1 continued on next page*

*Table 1 continued*

| Phylum | Species/strain | Complementation/recovery | | | | | Protein expression | | | | Native host characteristics | | Fe-S synthesis system |
|---|---|---|---|---|---|---|---|---|---|---|---|---|---|
| | | pNadA | pNadA + pBsSUF | pNadA + pEcSUF | pNadA −O2 | Complemented or recovered | Predicted translation initiation ($\log_{10}$) | SDS-PAGE | MS | Complemented/ recovered /detected | O₂ tolerance | Temperature | |
| | Bacillus subtilis subsp. subtilis str. 168 | – | + | + | + | + | 3.8 | – | N.T. | + | + | Mesophilic | SufBD |
| | Desulfitobacterium hafniense DCB-2 | – | – | – | + | + | 3.5 | – | + | + | – | Mesophilic | IscU |
| | Ruminiclostridium cellulolyticum H10 | – | – | – | – | – | 2.9 | – | N.T. | Detection N.T. | – | Mesophilic | IscU |
| Firmicutes | Heliobacterium modesticaldum Ice1 | – | – | – | – | – | 3.4 | – | + | + | – | Thermophilic | IscU |
| Nitrospinae | Nitrospina gracilis | + | N.T. | N.T. | + | + | 3.2 | + | N.T. | + | + | Mesophilic | SufBD |
| | Gemmata obscuriglobus | + | N.T. | N.T. | + | + | 2.1 | – | N.T. | + | + | Mesophilic | IscU |
| Planctomycetes | Pirellula staleyi DSM 6068 | – | – | – | – | – | 2.7 | – | + | + | + | Mesophilic | SufBD |
| | Aeromonas hydrophila | + | N.T. | N.T. | + | + | 3.8 | – | N.T. | + | + | Mesophilic | IscU |
| | Desulfovibrio vulgaris str. Hildenborough | – | – | – | – | – | 3.2 | – | N.T. | Detection N.T. | – | Mesophilic | SufBD, IscU |
| | Allochromatium vinosum | + | N.T. | N.T. | + | + | 2.9 | + | N.T. | + | – | Mesophilic | SufBD, IscU (NifU) |
| | Arcobacter butzleri | – | – | – | – | – | 2.2 | – | + | + | + | Mesophilic | SufBD, IscU |
| | Novosphingobium stygium | – | + | + | – | + | 2.3 | – | N.T. | + | + | Mesophilic | SufBD |
| | Herminiimonas arsenicoxydans | + | N.T. | N.T. | + | + | 3.5 | – | N.T. | + | + | Mesophilic | IscU |
| | Chromobacterium violaceum ATCC 12472 | + | N.T. | N.T. | + | + | 3.4 | – | N.T. | + | + | Mesophilic | IscU |
| | Bdellovibrio bacteriovorus HD100 | + | N.T. | N.T. | + | + | 3.9 | – | N.T. | + | + | Mesophilic | SufBD |
| | Methylobacillus flagellatus KT | – | – | – | – | – | 4.7 | – | + | + | – | Mesophilic | IscU |
| | Anaeromyxobacter dehalogenans 2 CP-C | + | N.T. | N.T. | + | + | 3.7 | – | N.T. | + | – | Mesophilic | SufBD, IscU |
| | Candidatus Pelagibacter ubique HTCC1002 | – | – | – | – | – | 3.4 | – | N.T. | Detection N.T. | + | ? | SufBD |
| | Syntrophobacter fumaroxidans MPOB | – | – | – | – | – | 3.5 | – | N.T. | Detection N.T. | – | Mesophilic | SufBD, IscU |
| | Cellvibrio japonicus Ueda107 | + | N.T. | N.T. | + | + | 2.9 | – | + | + | + | Mesophilic | IscU |
| | Escherichia coli str. K-12 substr. MG1655 | + | N.T. | N.T. | + | + | 3.8 | + | + | + | + | Mesophilic | SufBD, IscU |
| | Desulfarculus baarsii DSM 2075 | – | – | – | – | – | 2.8 | – | N.T. | Detection N.T. | – | Mesophilic | IscU |
| Proteobacteria | Salinisphaera sp. LB1 | + | N.T. | N.T. | + | + | 4.0 | – | N.T. | + | + | Mesophilic | SufBD |
| | Leptospira interrogans serovar Lai str. 56,601 | + | N.T. | N.T. | + | + | 4.2 | + | N.T. | + | + | Mesophilic | SufBD |
| Spirochaetes | Spirochaeta thermophila DSM 6578 | – | + | – | – | + | 3.2 | – | N.T. | + | – | Thermophilic | SufBD |
| Synergistetes | Thermanaerovibrio acidaminovorans DSM 6589 | – | – | – | – | – | N.T. | – | N.T. | Detection N.T. | – | Thermophilic | SufBD, IscU |
| Thermotogae | Thermotoga maritima MSB8 | – | – | – | + | + | 4.3 | – | N.T. | + | – | Hyperthermophilic | SufBD |
| Verrucomicrobia | Coraliomargarita akajimensis DSM 45221 | + | N.T. | N.T. | + | + | 3.6 | + | N.T. | + | + | Mesophilic | SufBD |

**Table 2.** Compiled results of complementation and protein expression experiments for each IspG ortholog.

Complementation of *Escherichia coli* Δ*ispG* and recovery of IspG activity with either coexpression of electron transfer proteins (pIspG + Fld or Fd) or anaerobic growth (pIspG −O$_2$). Electron transfer proteins found to activate each ortholog are listed in ***Supplementary files 5 and 6***. Minus (−), negative complementation; plus (+), positive complementation; N.T., not tested. Protein expression, predicted translation initiation rate from the RBS calculator (log$_{10}$ scale) and outcome of SDS-PAGE and mass spectroscopy detection experiments. SDS-PAGE column, (+) band observed on an SDS-PAGE gel. MS column, (+) confirmed detection by mass spectroscopy but no signal matching expected peptides; N.T., not tested; detection N.T., negative complementation and detection by MS not tested. The properties of native species for each ortholog are indicated: O$_2$ tolerance: (+) aerotolerant, (−) obligate anaerobe, (?) unknown; iron-sulfur (Fe-S) synthesis system describes homologs to IscU or SufB identified in native genome. SufBD, SufB homolog detected; IscU, IscU homolog detected; SufBC, IscU, both systems detected.

| Phylum | Species/strain | Complementation/recovery | | | | Protein expression | | | | Native host characteristics | | |
|---|---|---|---|---|---|---|---|---|---|---|---|---|
| | | pIspG | pIspG + Fld or fd | pIspG −O2 | Complemented or recovered | Predicted translation initiation rate (log$_{10}$) | SDS-PAGE | MS | Complemented/ recovered/ detected | O$_2$ tolerance | Temperature | Fe-S scaffold |
| Acidobacteria | *Candidatus Koribacter versatilis Ellin345* | − | N.T. | − | − | 3.6 | − | N.T. | Detection N.T. | + | Mesophilic | IscU |
| | *Thermoleophilum album* | − | N.T. | − | − | 2.9 | − | N.T. | Detection N.T. | + | Thermophilic | SufBD |
| | *Streptomyces coelicolor A3* | − | − | − | − | 2.6 | − | − | − | + | Mesophilic | SufBD |
| | *Rubrobacter xylanophilus DSM 9941* | + | N.T. | + | + | 2.6 | − | N.T. | + | + | Thermophilic | SufBD |
| | *Cutibacterium acnes KPA171202* | + | N.T. | + | + | 4.7 | − | N.T. | + | − | Mesophilic | SufBD |
| | *Collinsella stercoris DSM 13279* | − | − | + | + | 3.3 | − | − | + | − | Mesophilic | SufBD |
| | *Acidimicrobium ferrooxidans DSM 10331* | + | N.T. | + | + | 3.5 | − | N.T. | + | + | Mesophilic | SufBD |
| | *Streptomyces cattleya NRRL 8057 = DSM 46488* | − | + | − | + | 2.7 | + | N.T. | + | + | Mesophilic | SufBD |
| Actinobacteria | *Egibacter sp.* | − | N.T. | − | − | 3.4 | − | N.T. | Detection N.T. | ? | ? | SufBD |
| | *Aquifex aeolicus VF5* | + | N.T. | + | + | 3.8 | − | N.T. | + | + | Hyperthermophilic | IscU |
| Aquificae | *Thermovibrio ammonificans HB-1* | − | N.T. | − | − | 2.9 | − | N.T. | Detection N.T. | − | Thermophilic | SufBD |
| | *Pedobacter heparinus DSM 2366* | − | N.T. | − | − | N.T. | − | N.T. | Detection N.T. | + | Mesophilic | SufBD, IscU |
| Bacteroidetes | *Salinivirga cyanobacteriivorans* | − | N.T. | − | − | N.T. | − | N.T. | Detection N.T. | − | Mesophilic | SufBD |
| Candidatus Fermentibacteria | *Candidatus Fermentibacter daniensis* | − | N.T. | − | − | 3.0 | − | N.T. | Detection N.T. | − | Mesophilic | IscU |
| | *Candidatus Caenarcanum bioreactoricola* | − | N.T. | − | − | 4.1 | − | N.T. | Detection N.T. | ? | ? | IscU |
| Candidatus Melainabacteria | *Candidatus Gastranaerophilales bacterium HUM_7* | − | N.T. | − | − | 4.2 | − | N.T. | Detection N.T. | − | ? | IscU |
| Candidatus Sumerlaeota | *Candidatus Sumerlaea chitinovorans* | − | N.T. | − | − | 2.9 | − | N.T. | Detection N.T. | ? | ? | SufBD |
| Chlamydiae | *Chlamydia caviae GPIC* | − | − | − | − | 2.7 | − | − | − | + | Mesophilic | SufBD |

*Table 2 continued on next page*

Table 2 continued

| Phylum | Species/strain | Complementation/recovery | | | | Protein expression | | | | Native host characteristics | | |
|---|---|---|---|---|---|---|---|---|---|---|---|---|
| | | pIspG | pIspG + Fld or fd | pIspG –O2 | Complemented or recovered | Predicted translation initiation rate (log$_{10}$) | SDS-PAGE | MS | Complemented/ recovered/ detected | O$_2$ tolerance | Temperature | Fe-S scaffold |
| Chloroflexi | Sphaerobacter thermophilus DSM 20745 | – | N.T. | – | – | N.T. | – | N.T. | Detection N.T. | + | Thermophilic | SufBD |
| Cyanobacteria | Gloeobacter violaceus PCC 7421 | – | – | – | – | 3.6 | – | N.T. | Detection N.T. | + | ? | SufBD |
| | Synechocystis sp. PCC 6803 | – | + | – | + | 3.5 | – | N.T. | + | + | Mesophilic | SufBD |
| Deinococcus-Thermus | Thermus thermophilus HB27 | – | – | – | – | 2.7 | – | – | – | + | Thermophilic | SufBD |
| Fibrobacteres | Fibrobacter succinogenes subsp. succinogenes S85 | – | N.T. | – | – | 2.6 | – | N.T. | Detection N.T. | – | Mesophilic | SufBD |
| Firmicutes | Bacillus subtilis subsp. subtilis str. 168 | – | + | – | + | 2.7 | – | N.T. | + | + | Mesophilic | SufBD |
| | Desulfitobacterium hafniense DCB-2 | – | – | – | – | 3.4 | – | + | + | – | Mesophilic | SufBD |
| | Symbiobacterium thermophilum IAM 14863 | – | – | – | – | 2.0 | – | + | + | + | Thermophilic | SufBD |
| | Ruminiclostridium cellulolyticum H10 | – | – | – | – | 3.4 | – | + | + | – | Mesophilic | IscU |
| Lentisphaerae | Victivallales bacterium CCUG 44730 | – | N.T. | – | – | N.T. | – | N.T. | Detection N.T. | ? | ? | IscU |
| Nitrospinae | Nitrospina gracilis | – | – | – | – | 4.0 | – | – | – | + | Mesophilic | SufBD |
| | Blastopirellula marina DSM 3645 | +/– | + | + | + | 3.1 | – | N.T. | + | + | Mesophilic | SufBD |
| | Isosphaera pallida ATCC 43644 | – | – | – | – | 2.5 | – | + | + | + | Mesophilic | SufBD |
| Planctomycetes | Phycisphaera mikurensis NBRC 102666 | – | N.T. | – | – | 3.8 | – | N.T. | Detection N.T. | + | Mesophilic | SufBD |
| Proteobacteria | Aeromonas hydrophila | + | N.T. | + | + | 2.1 | – | N.T. | + | + | Mesophilic | IscU |
| | Allochromatium vinosum | – | – | – | – | 3.7 | – | + | + | – | Mesophilic | SufBD, IscU (NifU) |
| | Azoarcus sp. BH72 | + | N.T. | + | + | 3.1 | + | N.T. | + | + | Mesophilic | IscU |
| | Helicobacter pylori J99 | – | – | + | + | 3.3 | + | + | + | + | Mesophilic | IscU |
| | Methylococcus capsulatus str. Bath | + | N.T. | + | + | 3.2 | – | N.T. | + | + | Mesophilic | SufBD, IscU |
| | Rhodobacter sphaeroides 2.4.1 | – | + | – | + | 3.4 | – | N.T. | + | + | Mesophilic | SufBD, IscU (NifU) |
| | Mariprofundus ferrooxydans | – | + | – | + | 3.7 | – | + | + | + | Mesophilic | SufBD |
| | Cellvibrio japonicus Ueda107 | – | – | – | – | 2.6 | – | – | – | + | Mesophilic | SufBD |
| | Escherichia coli str. K-12 substr. MG1655 | + | N.T. | + | + | 3.1 | + | + | + | + | Mesophilic | SufBD, IscU |
| | Anaplasma phagocytophilum str. CRT38 | – | – | – | – | 2.2 | – | N.T. | Detection N.T. | + | ? | IscU |

Table 2 continued

| Phylum | Species/strain | Complementation/recovery | | | | Protein expression | | | | Native host characteristics | | Fe-S scaffold |
|---|---|---|---|---|---|---|---|---|---|---|---|---|
| | | plIspG | plIspG + Fld or fd | plIspG –O2 | Complemented or recovered | Predicted translation initiation rate (log$_{10}$) | SDS-PAGE | MS | Complemented/ recovered/ detected | O$_2$ tolerance | Temperature | |
| Spirochaetes | Spirochaeta thermophila DSM 6578 | – | – | – | – | 4.0 | – | + | + | – | Thermophilic | SufBD |
| Synergistetes | Thermanaerovibrio acidaminovorans DSM 6589 | – | N.T. | – | – | 3.1 | – | N.T. | Detection N.T. | – | Thermophilic | SufBD, IscU |
| Tenericutes | Spiroplasma citri | – | N.T. | – | – | 3.0 | – | N.T. | Detection N.T. | + | Mesophilic | SufBD |
| Thermodesulfobacteria | Thermodesulfatator indicus DSM 15286 | – | N.T. | – | – | 3.3 | – | N.T. | Detection N.T. | – | Thermophilic | SufBD |
| Thermotogae | Thermotoga maritima MSB8 | – | – | + | + | 4.4 | + | + | + | – | Hyperthermophilic | SufBD |
| Verrucomicrobia | Coraliomargarita akajimensis DSM 45221 | – | – | – | – | 2.6 | – | + | + | + | Mesophilic | SufBD |

of the *E. coli* SUF operon cloned into pBbA5a activated only three of the eight NadA orthologs activated by *B. subtilis* SUF (*B. subtilis*, *Mycolicibacterium smegmatis*, and *Novosphingobium stygium*) (*Table 1*). Thus, expression of heterologous Fe-S pathways broadens host compatibility with certain heterologous Fe-S enzymes. Interestingly, only NadA orthologs originating from organisms harboring SUF Fe-S biogenesis pathways could be recovered by *E. coli* or *B. subtilis* SUF expression (*Table 1*, *Figure 2E*).

## Phylogenetic comparison reveals oxygen sensitivity and Fe-S biogenesis pathway compatibility as two barriers that impede heterologous NadA activity

By mapping our complementation and recovery experiments onto NadA phylogeny, we could delineate two distinct clades (Clades 1 and 2) (*Figure 2A*). Twenty-two of the 23 heterologous orthologs within Clade 1 (which includes *E. coli* NadA) complemented growth of *E. coli* Δ*nadA* when expressed in aerobic conditions, of which 8 required coexpression with the *B. subtilis* SUF operon. In contrast, 5 of 24 Clade 2 orthologs could be recovered only by complementation in anaerobic conditions (*Figure 2A*). Only one Clade 2 ortholog (*B. subtilis* NadA) could be recovered in aerobic conditions with SUF overexpression. Interestingly, the majority of Clade 2 orthologs originate from organisms classified as obligate anaerobes (17/24) while most members of Clade 1 originate from aerotolerant organisms (17/23). Therefore, Clade 1 orthologs inactive in *E. coli* are limited by SUF expression or compatibility, whereas Clade 2 orthologs are likely limited by oxygen sensitivity. These results highlight oxygen sensitivity as the primary characteristic distinguishing Clade 1 and Clade 2 orthologs. Strikingly, membership within either clade predicts NadA activity in aerobic cultures more accurately than does the aerotolerance of the native hosts (*Figure 2E*).

## Compatibility of IspG orthologs is limited by need for taxa-specific electron transfer proteins

Coexpression of *E. coli* or *B. subtilis* SUF operons failed to restore activity of any IspG ortholog that did not complement *E. coli* Δ*ispG*. This suggests that steps facilitated by the Fe-S assembly pathway do not limit heterologous IspG activity. This was consistent with our observation that the *E. coli* Fe-S transfer protein ErpA delivers Fe-S clusters to heterologous IspG in vitro (*Figure 2—figure supplement 2*). The activity of *E. coli* IspG is known to require an electron carrier protein, a role performed in *E. coli* by flavodoxin 1 (FldA) (*Puan et al., 2005*). We therefore considered whether the inactivity of IspG orthologs is caused by incompatibility with *E. coli* electron carrier proteins. Although many enzymes that depend upon electron carrier proteins are able to accept a variety of carrier proteins (*Arcinas et al., 2019*; *Chazarreta-Cifre et al., 2011*), *E. coli* IspG specifically requires FldA (*Gaudu and Weiss, 2000*). Therefore we attempted to restore heterologous IspG activity using specific electron carrier proteins native to the organisms from which the IspG orthologs were obtained. Electron carriers that support activity of *Synechocystis* and *B. subtilis* IspG orthologs have been previously identified as the ferredoxin PetF (*Okada and Hase, 2005*) and flavodoxins YkuP and YkuN (*Kirby et al., 2016*), respectively. We therefore cloned and coexpressed *Synechocystis* PetF together with *Synechocystis* IspG within *E. coli* Δ*ispG*. Coexpression of PetF with *Synechocystis* IspG indeed complemented growth of *E. coli* Δ*ispG* (*Table 2*). Similarly, coexpression of either *B. subtilis* YkuP or YkuN with *B. subtilis* IspG complemented *E. coli* Δ*ispG*.

As the electron carriers needed to activate the remaining inactive IspG orthologs are not known, we first coexpressed synthetic multicistronic operons encoding electron carrier proteins and A-type Fe-S carrier proteins with a selection of IspG orthologs from the phylogenetically based set (*Supplementary file 5*). In total, five of the 12 IspG orthologs tested recovered activity (*Figure 2B and E*, *Table 2*, *Supplementary file 6*). Extending this approach to IspG orthologs used in the preliminary survey reactivated 3 out of 16 orthologs tested (*Supplementary file 6*). We next coexpressed individual electron carriers with IspG orthologs from *Rhodobacter sphaeroides*, *Synechocystis*, and *Streptomyces cattleya* to determine the specificity of the IspG orthologs. All three IspG orthologs were highly selective for specific electron carrier proteins: only one of the five electron carrier proteins tested from *R. sphaeroides* recovered activity of *R. sphaeroides* IspG. Only one of the two electron carriers from *Synechocystis* and two of the seven electron carriers tested from *S. cattleya* were able to activate their respective IspG orthologs (*Supplementary file 5*). In total, 5 out of 22 inactive IspG

orthologs tested from the phylogenetically driven dataset were reactivated by coexpression of electron carrier proteins. In all cases tested, expression of heterologous A-type carriers was not required to recover IspG activity.

We next explored the cross-species compatibility of IspG orthologs with non-native electron transfer proteins. With the exception of *B. subtilis* IspG, non-native electron transfer proteins did not activate any IspG ortholog tested in the phylogenetically based set. Among IspG orthologs used in the preliminary survey, far more cross-species activation by non-native electron transfer proteins was observed (*Supplementary file 6*). Specifically, *Synechocystis* PetF activated every *Cyanobacteria* IspG ortholog from this set. Ferredoxins from *S. cattleya* and *R. sphaeroides* activated additional IspG orthologs from the *Actinobacteria* and *Proteobacteria*, respectively. In total, 11 out of 27 inactive IspG orthologs tested from the preliminary dataset were reactivated by either native or heterologous electron transfer proteins (*Supplementary file 6*). Collectively, these results indicate that heterologous IspG can be recovered using compatible electron transfer proteins.

## Confirming heterologous expression of inactive orthologs

In order to determine whether any apparent inactivity of orthologs in our complementation assays is caused by inefficient translation, we used a quantitative biophysical model to predict translation rates of both NadA and IspG orthologs within the phylogenetically based set (*Salis et al., 2009*). Distributions of predicted initiation rates were obtained for ortholog groups organized by results from complementation and recovery experiments. No significant differences were observed among NadA and IspG groups (Kruskal-Wallis ANOVA) (*Figure 2—figure supplement 3*). Attempts to directly confirm expression of inactive orthologs using SDS-PAGE yielded mixed results, as even some of the orthologs that complemented growth could not be detected (*Tables 1 and 2* and *Supplementary file 7*). Focusing primarily upon orthologs that did not complement growth in any condition tested, we used a mass spectroscopy-based shotgun proteomics approach to confirm expression during aerobic growth without SUF or electron carrier protein coexpression. All seven NadA orthologs analyzed, including all five inactive orthologs, were detected. Of the 16 IspG orthologs analyzed, 10 were detected, including 8 of the 13 orthologs that could not be recovered (*Tables 1 and 2*, and *Supplementary file 7*, *Figure 2—figure supplement 3*).

As we cannot exclude that orthologs not analyzed or detected by mass spectroscopy were inefficiently translated, we compared the translation rate distributions obtained for these orthologs against orthologs that were detected or that could recover growth. No significant difference could be observed for NadA or IspG sets (*Figure 2—figure supplement 3*), suggesting that inefficient translation is not affecting our complementation assays. We also note that one ortholog (*Collinsella stercoris*, an obligate anaerobe), which required anaerobic conditions to complement growth of *E. coli* Δ*ispG*, could not be detected in the aerobic cultures prepared for mass spectroscopy analysis. As insoluble proteins may evade detection by mass spectroscopy, and as overexpressed *E. coli* IspG is prone to insolubility (*Zhou et al., 2012*), we speculate that insolubility may prevent detection and activity of these orthologs.

## In vitro detection of Fe-S clusters in purified NadA and IspG orthologs

To determine whether orthologs that are inactive during aerobic growth obtain Fe-S clusters, we expressed NadA orthologs from *Dictyoglomus thermophilum* and *Desulfurispirillum indicum* and the IspG ortholog from *Thermotoga maritima* within aerobic *E. coli* cultures. Following anaerobic purification, UV-visible spectroscopy and quantification of iron and sulfur confirmed the presence of Fe-S clusters within a small fraction of each purified ortholog (*Figure 2—figure supplement 4*). As each of these orthologs complemented growth in anaerobic cultures, inactivity during aerobic conditions suggests that the Fe-S clusters detected in the purified enzymes may be oxidized or otherwise inactivated. Surprisingly, an Fe-S cluster was also detected in a significant fraction of purified *Symbiobacterium thermophilum* IspG, an ortholog that could not complement growth of *E. coli* Δ*ispG* in any condition tested. This intriguing observation suggests that *S. thermophilium* IspG requires a specific electron transfer protein for enzyme activity, but not for acquiring its Fe-S cluster.

## A specific double-cluster ferredoxin enhances the activity of the Fe-S-dependent methyltransferase TsrM

In order to better understand why Fe-S enzymes require electron transfer proteins with specific properties, we explored the electron transfer protein selectivity of a biotechnologically relevant Fe-S enzyme (*Broderick et al., 2014*). rSAM methyltransferase enzymes rely upon Fe-S clusters to catalyze biosynthesis of valuable natural products, including antibiotics (*Mahanta et al., 2017*; *Mehta et al., 2015*). The cobalamin-dependent rSAM methyltransferase TsrM, an enzyme from the thiostrepton synthesis pathway of *Streptomyces laurentii*, uses its cobalamin cofactor as an intermediate carrier to transfer a methyl group to C2 of tryptophan (*Blaszczyk et al., 2016*). Although its catalytic cycle does not require an electron donor for each turnover event, TsrM requires one electron to reduce its cobalamin cofactor to cob(I)alamin, which reacts with SAM to form MeCbl (*Figure 3A*). TsrM can also be inactivated by adventitious oxidation of the bound cobalamin factor from cob(I)alamin to cob(II)alamin. Recovering the catalytically active cob(I)alamin form requires reduction by an electron transfer protein (*Blaszczyk et al., 2019*).

*E. coli* NCM3722 coexpressing *S. laurentii* TsrM together with the *E. coli* cobalamin import pathway (*btu* operon expressed from a plasmid [*Lanz et al., 2018*]) produced 3 ± 2 mg/L 2-methyltryptophan (*Figure 3B*). We sought to determine whether in vivo TsrM activity could be improved using *Streptomyces* [4Fe-4S] ferredoxins, as [4Fe-4S] ferredoxins exhibit low reducing potentials (–700 to –300 mV) (*Cammack, 1992*) that are expected to match the low reduction potential of cob(II)alamin/cob(I)alamin (<–500 mV vs NHE) (*Jarrett et al., 1997*; *Lexa and Saveant, 2002*). We used three [4Fe-4S] ferredoxins (here designated Fdx1, Fdx2, and Fdx3), two of which (Fdx1 and Fdx3) were able to activate *S. cattleya* IspG (*Supplementary file 5*). Simultaneous coexpression of all three [4Fe-4S] ferredoxins together with TsrM increased 2-methyltryptophan titer to 36 ± 16 mg/L. By coexpressing each ferredoxin individually with TsrM, we determined that Fdx1 alone improves activity of TsrM (45 ± 7 mg/L 2-methyltryptophan), while expression of Fdx2 and Fdx3 does not increase 2-methyltryptophan titers (*Figure 3B*).

We further investigated the molecular basis of ferredoxin selectivity by TsrM using in vitro experiments with purified proteins. Consistent with previous observations, purified TsrM contains 0.9 cobalamin and 3.5 Fe per monomer (*Figure 3—figure supplement 1*). The iron content of the ferredoxins after anaerobic purification or reconstitution with iron and sulfur salts (7.4, 2.9, and 2.8 Fe atoms *per* monomer for Fdx1, Fdx2, and Fdx3, respectively) is consistent with their corresponding annotations as double-cluster and single-cluster ferredoxins. We first used biolayer interferometry (BLI) to determine whether TsrM associates preferentially with Fdx1. While we were able to detect association between Fdx1 and TsrM (dissociation constant $K_D$ = 40 µM), we found that Fdx2 and Fdx3 did not measurably associate with TsrM (*Figure 3C*). We also compared the redox potentials of each ferredoxin against TsrM using voltammetry. The reduction potentials determined for the [4Fe-4S] cluster of TsrM match with cob(II)alamin/cob(I)alamin (–730 mV vs Ag/AgCl), in agreement with previous reports (*Blaszczyk et al., 2016*). The voltammogram of Fdx1 indicated two redox transitions associated with each [4Fe-4S] cluster (–750 mV and –650 mV vs Ag/AgCl), the lower of which supports reduction of TsrM metallic cofactors ([4Fe-4S]$^{2+}$ to [4Fe-4S]$^{1+}$ cluster) and (cob(II)alamin to cob(I)alamin). The redox potentials of [4Fe-4S] Fdx2 and [4Fe-4S] Fdx3 are each too high for favorable reduction of TsrM metallic cofactors (–620 mV and –585 mV respectively vs Ag/AgCl) (*Figure 3D*). Therefore, a lack of electron transfer proteins that exhibit both binding specificity with TsrM and suitably low reduction potential constrain the functional heterologous expression of TsrM.

## Discussion

Fe-S clusters are ancient metal cofactors present in all domains of life and genes encoding Fe-S containing proteins are likely to be transferred between organisms. Moreover, a wide array of Fe-S enzymes have biotechnological importance. However, inefficient maturation in heterologous hosts may impede activity of Fe-S proteins acquired via lateral transfer. By systematically mapping the transferability of Fe-S enzyme activities between prokaryote species, we have identified several factors affecting heterologous Fe-S enzyme activity: oxygen sensitivity, incompatibility with the host Fe-S biogenesis network, and incompatibility with host electron transfer proteins. Many orthologs are likely to be limited by multiple factors, e.g., oxygen sensitivity and Fe-S network incompatibility. The

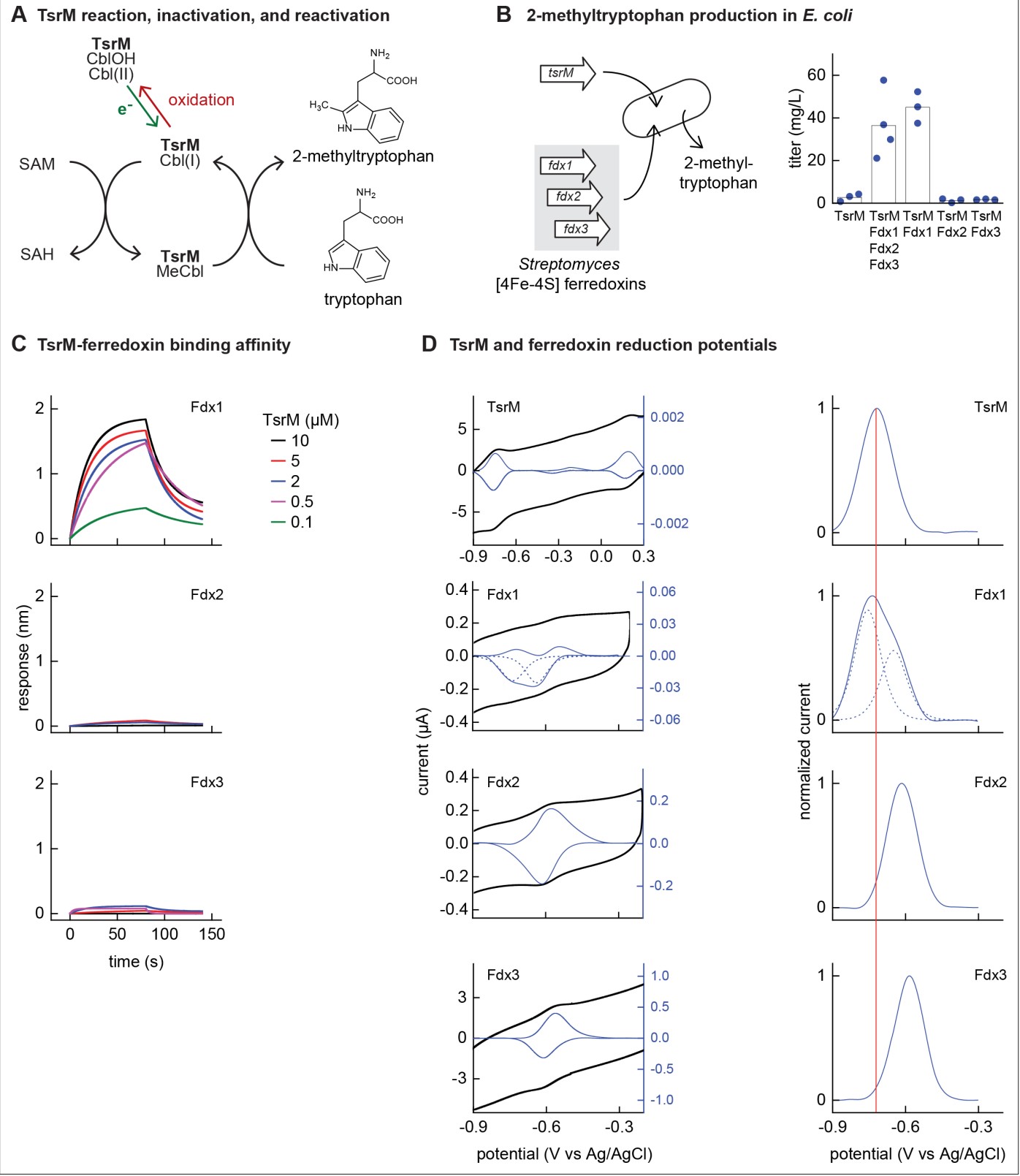

**Figure 3.** Identifying the barrier to optimal activity of an iron-sulfur (Fe-S)-dependent rSAM enzyme. (**A**) The enzyme TsrM synthesizes 2-methyltryptophan as part of the synthesis pathway of the antibiotic thiostrepton. A one-electron reduction (in green) converts the bound cobalamin to cob(I)alamin, which is converted to MeCbl via reaction with SAM. TsrM can be inactivated during catalysis by adventitious oxidation of the bound cob(I)alamin to cob(II)alamin (in red). Recovering the catalytically active cob(I)alamin form requires a one-electron reduction (green). (**B**) Titers of

*Figure 3 continued on next page*

*Figure 3 continued*

2-methyltryptophan in a TsrM-expressing *E. coli* strain increase substantially upon coexpression of a specific *Streptomyces cattleya* ferredoxin (Fdx1) whereas coexpression of alternative *S. cattleya* [4Fe-4S] ferredoxins, Fdx2 and Fdx3, does not increase titers. (**C, D**) Identifying the molecular basis enabling Fdx1 to recover TsrM activity. (**C**) Biolayer interferometry of TsrM with Fdx1, Fdx2, and Fdx3 indicates that TsrM interacts measurably with Fdx1 alone. (**D**) Cyclic voltammetry of TsrM and the three ferredoxins indicates that the reducing potentials of one of the (Fe-S) clusters of Fdx1 are sufficiently low potential to reduce TsrM metal cofactors (cob(II)alamin to cob(I)alamin and [4Fe-4S]$^{+2}$ to [4Fe-4S]$^{+1}$) and thus restore activity after adventitious oxidation.

The online version of this article includes the following figure supplement(s) for figure 3:

**Figure supplement 1.** TsrM, Fdx1, Fdx2, and Fdx3 with their metallic cofactors.

adaptations required for Fe-S enzymes to maintain activity within aerobic organisms are reflected by the existence of separate aerotolerant and oxygen-sensitive clades in NadA phylogeny. This in turn leads to a remarkable correlation between the likelihood of complementation by NadA orthologs and phylogenetic distance from *E. coli* NadA. Structural studies, as have been performed for the Fe-S enzyme IspH (*Rao and Oldfield, 2016*), could further explore the structural mechanisms underlying the oxygen tolerance of NadA orthologs from Clade 1.

Many NadA orthologs within the aerotolerant clade required a heterologous Fe-S biogenesis pathway, indicating that incompatibility with *E. coli* Fe-S biogenesis pathways can impede heterologous activity. Interestingly, NadA orthologs reactivated by SUF coexpression were obtained entirely from organisms that featured only an SUF pathway, suggesting that some orthologs require specific Fe-S scaffold or carrier proteins. Nevertheless, the native *E. coli* Fe-S cluster transfer and maturation pathways proved impressively versatile with heterologous Fe-S enzymes, as more than one-third of NadA orthologs tested – and by extension from our phylogenetically representative set, the same fraction of all known prokaryotic NadA orthologs – obtain Fe-S clusters when expressed within *E. coli*. Our results with IspG orthologs also reflect the versatility of the *E. coli* Fe-S network, as many IspG orthologs were reactivated using electron transfer proteins alone. For such orthologs, IspG activity is not limited by incompatibilities with any step facilitated by host Fe-S biogenesis pathways. Although Fe-S pathways must be versatile to provide Fe-S clusters to many structurally and functionally diverse Fe-S enzymes within a single species (e.g. >150 Fe-S enzymes in *E. coli*), it is nevertheless remarkable that this versatility also extends to Fe-S enzymes from distantly related species.

Unlike NadA, IspG orthologs at even a low patristic distance from *E. coli* IspG were inactive if compatible electron transfer proteins were not provided. The electron transfer proteins we identified may contribute essential roles during Fe-S cluster maturation, enzymatic catalysis, or cluster reduction following adventitious oxidation. The finding that electron transfer specificity prevents functional expression to such a severe degree was unexpected as many redox-dependent enzymes are compatible with multiple electron transfer proteins and vice versa. For instance, the *E. coli* flavodoxin FldA delivers electrons to *E. coli* BioB, IspG, and IspH (*Wolff et al., 2003*). Conversely, three *B. subtilis* electron transfer proteins (including YkuN and YkuP) support activity of *B. subtilis* acyl lipid desaturase (*Chazarreta-Cifre et al., 2011*) and five *T. maritima* ferredoxins are compatible with the *T. maritima* rSAM enzyme MiaB (*Arcinas et al., 2019*). Despite this promiscuity, prokaryotic genomes retain multiple electron transfer proteins whose structures and reduction potentials vary considerably. *E. coli* IspG requires low-potential reductants ($E' < -300$ mV) (*Xiao et al., 2009*), suggesting that FldA reduces IspG as a hydroquinone ($E' = -433$ mV) (*McIver et al., 1998*). However, the FldA potential is well below the corresponding potentials of *B. subtilis* YkuP and YkuN ($-377$ and $-382$ mV, respectively) as well as the reduction potential of cyanobacterial ferredoxins similar to *Synechocystis* PetF (approximately $-380$ mV) (*Lawson et al., 2004*; *Motomura et al., 2019*). Assuming that IspG orthologs exhibit reduction potentials similar to *E. coli* IspG, the selectivity we observe is likely determined by structural complementarity between electron transfer proteins and IspG orthologs. The presence of multiple electron transfer proteins enables specialization (*Kaçar and Gaucher, 2013*), and the retention of multiple electron transfer proteins reflects the nonredundant roles they play in their host. While this specialization may extend biosynthetic capabilities, it may also constrain the compatibility of electron transfer-dependent enzymes with heterologous hosts.

Fe-S enzymes catalyze many reactions that are relevant to biotechnology such as nitrogen reduction, dehydration, sulfur insertion, and methyl transfer (*Johnson et al., 2005*). The full potential of Fe-S enzymes cannot be harnessed until we are able to routinely express them within heterologous species.

By identifying the separate contributions of oxygen sensitivity, Fe-S biogenesis network compatibility, and electron transfer protein specificity to impeding Fe-S enzyme function, our complementation and recovery assays suggest guidelines for improving the compatibility of Fe-S enzymes with engineered organisms. First, it should be explored whether identifying distinct phylogenetic clades within Fe-S enzyme families may reveal orthologs found predominantly in aerobic organisms, as we observed for NadA. Such efforts could guide selection of heterologous enzymes compatible with aerobic conditions. As we found with NadA, enzyme phylogeny may be more predictive of heterologous activity in aerobic conditions than the aerotolerance of the host organism. Second, the common approach of overexpressing host Fe-S pathways may not prove sufficient for some Fe-S enzymes. Many Fe-S enzymes may instead require heterologous Fe-S pathways (*Corless et al., 2020*; *Nakamura et al., 1999*). Our results with *B. subtilis* SUF suggest that the heterologous pathway used should match that of the original host. Finally, many heterologous Fe-S enzymes are often inactive not due to incompatibilities with Fe-S networks, but rather a lack of other essential compatible proteins such as electron transfer proteins, the importance of which may be often overlooked. As we demonstrated with IspG, suitable cofactors from the genome of the original host can often be identified by coexpression with the heterologous Fe-S enzyme. Likewise, in the case of TsrM, the low reducing potential of cobalamin informed the selection of [4Fe-4S]-type ferredoxins for screening. Additional studies evaluating these guidelines with a wider array of enzyme families could help formulate 'design rules' enabling the routine use of Fe-S enzymes in biosynthetic pathways, thus unlocking the chemical versatility of Fe-S clusters.

## Materials and methods
### Selection of sequences and bioinformatics analysis

Orthologs of IspG, NadA, and IspD selected in the preliminary screen were found in a variety of sequenced prokaryotic genomes using homology searches guided by *E. coli* sequences. To assemble the phylogenetically representative distribution of NadA and IspG orthologs that was assembled following the preliminary screen, a systematic approach was used. First, a protein database was built from 248 representative prokaryotic proteomes (*Supplementary file 2*) (including 65 proteomes that were used for preliminary screen) gathered from NCBI FTP (ftp://ftp.ncbi.nlm.nih.gov/genomes/all), selecting around one genome *per* order (27 archaea and 221 bacteria) by using NCBI taxonomy and privileging complete proteomes. Homologs of NadA and IspG were first identified by BLASTp v2.8.1+ (*Altschul et al., 1997*), using *E. coli* sequences as seeds (AAC73837.1 and AAC75568.1, respectively). Sequences associated to an *E* value lower than $10^{-4}$ were aligned with MAFFT v7.419 (*Katoh and Standley, 2013*) and used to build hidden Markov model (HMM) profiles using hmmbuild from the HMMER v3.2.1 package (*McClure et al., 1996*). These HMM profiles were then used to request the database using hmmsearch, and the sequences presenting an *E* value lower than $10^{-2}$ were retrieved and aligned. Alignments were manually curated using Aliview v1.25 (*Larsson, 2014*) and trimmed using BMGE v1.1 (*Criscuolo and Gribaldo, 2010*) with the substitution matrix BLOSUM30.

The best suited model for each alignment was selected using ModelFinder implemented in IQ-TREE v1.6.10 (*Nguyen et al., 2015*), according to the Bayesian Information Criteria (LG + I + G4 for both NadA and IspG). Maximum Likelihood trees were inferred using PhyML v3.1 (*Guindon et al., 2010*) and 100 bootstrap replicates performed to assess the robustness of the branches by using the transfer bootstrap value using BOOSTER v1.0 (*Lemoine et al., 2018*). For both NadA and IspG, the number of copies *per* genome was in the extreme majority equal to 1. All homologs are thus considered as orthologs.

In order to select representative NadA and IspG sequences to be experimentally tested, the ML trees were split into clusters using TreeCluster v1.0 (*Balaban et al., 2019*) with a threshold of 1.75, leading to 15 sequence clusters for NadA and 17 for IspG. Then, each cluster was further split into subclusters using hierarchical clustering implemented in scipy.cluster.hierarchy python library (*Virtanen et al., 2020*). The number of subclusters for each cluster was set to 20% of the sequences within each cluster. One sequence per subcluster was finally selected, privileging reference strains and already tested sequences, leading to 47 sequences for NadA and 47 for IspG. The representativeness was checked by comparing the shape of distribution of patristic distances between the entire NadA and IspG trees and the selected sequences. The correlation between complementation data

and phylogenetic distance was calculated by the correlation ratio $\eta^2$. Data for phylogenetic analysis (containing protein database of the 248 prokaryotes, NadA/IspG alignments and phylogenies and python script used to select sequences by clustering approach) is available at figshare.com/articles/dataset/Dangelo_et_al_Supplementary_data_zip/13664927.

The native host characteristics have been found using BacDive (*Reimer et al., 2019*) and by a curation of literature. The Fe-S cluster biogenesis systems have been identified by doing a BLASTp v2.8.1+ on the 248 proteome database, starting from IscU and SufB from *E. coli*. The genomes containing homologs of SufB were considered as possessing SufB-based system. Then, IscU-based system has been considered when encoding an IscU homolog not in context with SufBD (SufU) and in context with a cysteine desulfurase.

## Bacterial strains, media, and chemicals

The plasmids, oligonucleotides, and *E. coli* strains used in this study are listed in *Supplementary file 8*. *E. coli* Δ*ispG* was constructed by first integrating genes encoding the lower half of the mevalonate (MVA) pathway to enable isoprenoid synthesis from exogenously provided MVA. Genes encoding mevalonate kinase, phosphomevalonate kinase, and mevalonate diphosphate decarboxylase from *Saccharomyces cerevisiae* control by the $P_{trc}$ promoter was amplified from pJBEI-2999 (*Peralta-Yahya et al., 2011*) and assembled by PCR to an FRT-flanked kanamycin resistance (KanR) cassette from pKD13 and integrated into *intA* using homologous recombination (*Datsenko and Wanner, 2000*). The KanR cassette was subsequently removed using the plasmid pCP20. *nadA,* was removed using homologous recombination in *E. coli* MG1655. Bacterial strains were routinely grown in aeration at 37°C in Luria-Bertani broth (LB) or in M9 medium (M9) supplemented with glucose (0.4%), $CaCl_2$ (100 μM), and $MgSO_4$ (1 mM). Solid media contained 1.5% agar. Anhydrotetracycline (aTc) 100 ng/mL, isopropyl β-D-1-thiogalactopyranoside (IPTG) 250 μM, MVA 0.5 mM, and nicotinic acid (NA) 10 μg/mL were added when indicated. When required, antibiotics were added at the concentrations of 50 μg/ml kanamycin (Kan) and 100 μg/mL ampicillin (Amp). Mevalolactone (MVL) and NA were purchased from Sigma-Aldrich and resuspended in water at final concentrations of 1 M and 10 mg/mL, respectively. To prepare MVA, an equal volume of 1 M KOH was added to 1 M MVL and incubated at 37°C for 30 min.

## Plasmid construction for complementation assays

All the genes encoding heterologous orthologs and electron carrier proteins (with the exception of genes from *B. subtilis*) were codon optimized for expression in *E. coli*, designed with strong ribosome binding sites, synthesized, and cloned into the expression vector pBbS2k (*Lee et al., 2011*) by the Joint Genome Institute (JGI) directly, or were obtained as gene fragments by Twist Bioscience (TWB) (*Supplementary files 1 and 4*), assembled, and cloned into pBbS2k. DH5α was used for all cloning steps. Genes encoding electron carrier proteins were codon optimized for expression in *E. coli*, designed with strong ribosome binding sites, and assembled into the expression vector pBbA5a by JGI.

## Complementation assays

*E. coli* knockout strains were transformed with pBbS2k plasmid encoding corresponding orthologs and selected on permissive LB medium containing kanamycin. Plates supporting growth of *E. coli* Δ*ispG* strains additionally contained 0.5 mM MVA. For plate-based complementation assays, kanamycin-resistant colonies were streaked on LB plates supplemented with kanamycin and aTc (for IspG experiments) or on solid M9 glucose medium supplemented with kanamycin and aTc (for NadA experiments). Plates were incubated at 37°C and growth was scored after 16 and 40 hr. For anaerobic experiments, plates were incubated inside an anaerobic box (AnaeroPack). For liquid culture-based complementation experiments, two individual colonies for each strain were inoculated into LB medium, grown for 7 hr at 37°C, and subsequently diluted 1:500 in M9 medium (LB for IspG experiments) containing Kan and aTc. For reactivation experiments of inactive NadA orthologs, an *E. coli* Δ*nadA* strain bearing a plasmid encoding the *B. subtilis* SUF operon (pBbA5a-*sufCDSUB*) or the plasmid encoding the *E. coli* SUF operon (pBbA5a-*sufABCDSE*) was transformed with plasmids encoding orthologs and grown as described with the addition of 250 μM IPTG to induce expression of SUF (*Supplementary files 5 and 8*). For reactivation experiments of inactive IspG orthologs, an *E. coli* Δ*ispG* strain bearing a plasmid encoding IspG orthologs was transformed with pBbA5a plasmids encoding electron transfer proteins.

All growth curves have been performed in 96-wells microplates and cell density (OD$_{600}$) was recorded by using an automated Spark 10Mluminometer-spectrophotometer (Tecan) for 18 hr (NadA experiments) or 12 hr (IspG experiments), every 30 min at 37°C in shaking condition.

## SDS-PAGE test for protein expression

Δ*nadA* strains containing the pBbS2k plasmids and carrying NadA orthologs were grown in LB, Kan, and aTc at 37°C to OD$_{600}$ of 1.2–1.5. Δ*ispG* strains containing the pBbS2k plasmids carrying the IspG orthologs were grown in LB, Kan, MVA, and aTc at 37°C to OD$_{600}$ of 1.2–1.5. In both cases, the cells were centrifuged at 12,000 rpm for 2 min at room temperature and after removing the supernatant, the pellets were resuspended in PBS 1× buffer and then loaded on 12% SDS-PAGE gel after denaturating with Leammli loading dye at 100°C for 15 min. The gel was stained with Coomassie brilliant blue G-250 and decolorized using a solution of 60% water, 30% isopropanol, and 10% acetic acid.

## Mass spectrometry detection of orthologs

### Sample preparation

Δ*ispG* and Δ*nadA* strains bearing corresponding inactive orthologs were inoculated in 10 mL permissive media (Δ*ispG* in LB with 0.5 mM MVA, 100 µM IPTG, 25 µg/mL kanamycin; or Δ*nadA* in LB with 25 µg/mL kanamycin). When cultures reached early exponential phase (OD = 0.1), ortholog expression was maximally induced by 10 ng/mL doxycycline (added at early exponential phase before OD = 0.1). The cultures were subsequently grown to midlogarithmic phase (OD = 0.4–0.6), at which point 2 mL solution of 10% (w/vol) trichloroacetic acid (TCA) was added. Quenched samples were incubated on ice for at least 10 min before centrifugation (20,000 *g*, 4°C, 10 min), after which the supernatant was removed and pellet stored at –80°C before analysis.

### Cell lysis and protein extraction

Briefly, biomass amount equivalent to 1 mL of 1 OD *E. coli* was collected in an Eppendorf tube and solubilized in a suspension solution consisting of 200 µL B-PER reagent (Thermo Scientific) and 200 µL TEAB buffer (50 mM triethylammonium bicarbonate (TEAB), 1% (w/w) NaDOC, adjusted to pH 8.0) including 0.2 µL protease inhibitor (P8215, Sigma Aldrich). Further, 0.1 g of glass beads (acid, washed, approx. 100 µm diameter) were added and cells were disrupted using three cycles of bead beating on a vortex for 30 s followed by cooling on ice for 30 s in-between cycles. In the following, a freeze/thaw step was performed by freezing the suspension at –80°C for 15 min and thawing under shaking at elevated temperature using an incubator. The cell debris was further pelleted by centrifugation using a bench top centrifuge at max speed, under cooling for 10 min. The supernatant was transferred to a new Eppendorf tube and kept at 4°C until further processed. Protein was precipitated by adding one volume of TCA ( Sigma Aldrich) to four volumes of supernatant. The solution was incubated at 4°C for 10 min and further pelleted at 14,000 *g* for 10 min. The obtained protein precipitate was washed twice using 250 µL ice cold acetone.

### Proteolytic digestion

The protein pellet was dissolved to approx. 100 µg/100 µL of 200 mM ammonium bicarbonate containing 6 M urea to a final concentration of approximately 100 µg/µL. To 100 µL protein solution, 30 µL of a 10 mM dithiothreitol (DTT) solution were added and incubated at 37°C for 1 hr. In the following, 30 µL of a freshly prepared 20 mM iodoacetic acid (IAA) solution was added and incubated in the dark for 30 min. The solution was diluted to below 1 M urea using 200 mM bicarbonate buffer and an aliquot of approximately 25 µg protein was digested using sequencing grade trypsin at 37°C overnight (trypsin/protein 1:50). Finally, protein digests were then further desalted using an Oasis HLB 96 well plate (waters) according to the manufacturer protocols. The purified peptide eluate was dried using a speed vac concentrator.

### One-dimensional shot gun proteomics approach

Briefly, the samples were analyzed using a nanoliquid chromatography system consisting of an ESAY nano LC 1200, equipped with an Acclaim PepMap RSLC RP C18 separation column (50 µm × 150 mm, 2 µm), and an QE plus Orbitrap mass spectrometer (Thermo). The flow rate was maintained at 350 nL/min over a linear gradient from 4% to 30% solvent B over 32.5 min, and finally to 70% B

over 12.5 min. Solvent A was $H_2O$ containing 0.1% formic acid, and solvent B consisted of 80% acetonitrile in $H_2O$ and 0.1% formic acid. The Orbitrap was operated in data-depended acquisition mode acquiring peptide signals form 385 to 1250 m/z at 70 K resolution with a max IT of 100 ms and an AGC target of 3e6. The top 10 signals were isolated at a window of 2.0 m/z and fragmented using an NCE of 28. Fragments were acquired at 17 K resolution with a max IT of 75 ms and an AGC target of 2e5. Singly charged, 6× and higher charged mass peaks were excluded from selection. Data were acquired from 0 to 60 min.

### Database search

Data were analyzed against the proteome database from *E. coli* K-12 (UniprotKB, TaxID 83333), including amino acid sequences for heterologous orthologs, using PEAKS Studio 10.0 (Bioinformatics Solutions Inc) allowing for 20 ppm parent ion and 0.02 m/z fragment ion mass error, two missed cleavages, carbamidomethylation as fixed and methionine oxidation and N/Q deamidation as variable modifications. Peptide spectrum matches were filtered against 1% false discovery rate and protein identifications with ≥2 unique peptides were accepted as significant.

## 2-Methyl tryptophan biosynthesis

The mature protein coding sequence of the tryptophan 2-C-methyltransferase (TsrM) from *S. laurentii* (GenBank: FJ652572.1) was codon optimized for *E. coli* expression and synthetized by TWB. The optimized TsrM sequence was inserted into a pBbA5k BglBrick backbone between the BglII and BamHI sites to obtain the vector pTsrM. Codon-optimized genes encoding electron transfer proteins were cloned between the BamHI and XhoI sites of the pTsrM vector (**Supplementary file 8**). The vector for cobalamin importer overexpression was constructed similarly to a previously published report (**Lanz et al., 2018**). The genes for proteins BtuB (AYG21241.1, GenBank), BtuC (AYG19236.1, GenBank), BtuD (AYG19238.1, GenBank), BtuE (AYG19237.1, GenBank), and BtuF (AYG20683.1, GenBank) were amplified from *E. coli* MG1655 genome and cloned into the backbone pBbS2c BglBrick backbone to create the vector pBtu.

2-Methyl tryptophan was produced using *E. coli* NCM3722 pBtu and with plasmids pTsrM, pTsrM_Scatt3Fd, pTsrM_Fdx1, pTsrM_Fdx2, or pTsrM_Fdx3. Individual colonies were picked and inoculated for precultures grown overnight in 1 mL of SAM MOPS minimal medium (MOPS minimal medium [**Neidhardt et al., 1974**] supplemented with 1% w/vol glucose, 0.25% Casamino acids, and 7 μM hydroxocobalamin), at 37°C. The cultures (triplicates) were obtained by inoculating 50 μL of precultures in 3 mL of SAM MOPS minimal medium. The cultures were grown at 37°C to $OD_{600}$ of 0.05–0.1 for pBtu induction (83.3 ng/mL aTc) and continued to grow in the same conditions. At $OD_{600}$ 0.5, TsrM and ferredoxin expression were induced (0.25 mM IPTG) and the medium supplemented with 0.15 mM cysteine and 32.5 μM $FeCl_3$. Cultures were incubated at room temperature. Samples were taken after 24 hr (1 ml and the equivalent of 1 ml sample with 0.5 $OD_{600}$ samples). Samples were centrifuged at 15,000 rpm for 2 min, and supernatants were transferred to new tubes. Pellets were quenched with quenching solution (methanol, acetonitrile, and water, ratio of 5:3:1) + 0.1% formic acid and resuspended. Pellet samples were dried in speedvac at 40°C and resuspended with quenching solution for LC-MS measurement.

## 2-Methyltryptophan quantification

2-Methyl tryptophan was quantified by an LC-MS system (Agilent) using an XBridge BEH Amide 2.5 μm (Waters, Bridge Columns) with a precolumn and equipped with a standard ESI source mass spectrometer (sample injection volume of 5 μL). The mobile phase was compound by two solvents, A (20 mM ammonium formate in 10% acetonitrile) and B (20 mM ammonium formate in 80% acetonitrile). After 6 min at 100% solvent B, the metabolites were separated by a gradient from 100% to 70% of solvent B for 6 min (flow rate 0.4 mL/min), followed by a gradient from 70% to 100% for 50 s (same flow rate) and held at 100% solvent B for 3 min 10 s. The 2-methyl tryptophan precursor ion (positive polarity, 219.1) was fragmented into product ions (144.1 and 128) using an ESI ionization in MRM mode. 2-Methyl tryptophan concentration was estimated using a calibration curve constructed with standard samples.

## Expression and purification of heterologous IspGs

Coding sequences of all tested IspG proteins were ordered codon optimized and cloned in a pET28a(+) plasmid by Genscript. A tobacco etch virus (TEV) protease cleavage site was inserted between the N-Terminal His-tag and the downstream IspG sequences. Each IspG was expressed and purified under aerobic conditions. Expression was carried out in BL21(DE3) cells for 3 hr under 1 mM IPTG induction at 37°C. Harvested cells were resuspended in lysis buffer (50 mM Tris pH 7.5, 250 mM NaCl, 10 mM imidazole), sonicated on ice (5 min, 10/30 s ON/OFF, power 70%), and resulting lysate was clarified by ultracentrifugation (40,000 $g$, 20 min, 4°C). Supernatant was loaded onto a 5 mL Ni-NTA column. Column was washed using lysis buffer +20 mM imidazole and proteins were eluted with lysis buffer +300 mM imidazole. Fractions containing IspG were pooled together and incubated with TEV protease harboring a His-tag (molar ratio: 1/100) o/n at 4°C for dialysis against storage buffer (50 mM Tris pH 7.5, 250 mM NaCl, 1 mM DTT). Dialysis bag content was reloaded onto 5 mL Ni-NTA column and flow-through (FT) containing IspG (without His-tag) was collected. Pure apo-IspG was concentrated and stored at –80°C. To counter instability of *E. coli* IspG, glycerol (50% final) was added to concentrated purified protein which was subsequently stored at –20°C.

## Anaerobic purification of IspG and NadA orthologs

NadA orthologs from *D. thermophilum* and *D. indicum* and IspG orthologs from *T. maritima* and *S. thermophilum* were introduced for protein expression into pET22 and pET6H, respectively (**Supplementary file 8**). Expression was carried out aerobically in MG1655(DE3) cells for 3 hr under 0.5 mM IPTG induction at 37°C. Purifications of all proteins were performed the same day under anaerobiosis inside a glove box (Jacomex, $O_2$ < 2 ppm) in lysis buffer (100 mM Tris pH 7.5, 150 mM NaCl), sonicated (10 min, 10/30 s ON/OFF, power 70%) and resulting lysate was clarified by ultracentrifugation (40,000 $g$, 20 min, 4°C). Supernatant was loaded anaerobically onto a 5 mL Ni-NTA column. Column was washed using lysis buffer +20 mM imidazole and proteins were eluted with lysis buffer +300 mM imidazole. Fractions containing enzymes (IspG and NadA) were desalted to remove imidazole using NAP-25 columns in 100 mM Tris-HCl pH 8, 150 mM NaCl and analyzed by UV-visible absorption spectroscopy and for their iron and sulfur content (*Beinert, 1983*; *Fish, 1988*).

## Expression, purification, and reconstitution of *E. coli* ErpA

*E. coli* ErpA was obtained as previously reported (*Loiseau et al., 2007*).

## Fe-S cluster transfer between *E. coli* ErpA and heterologous IspGs

Fe-S cluster transfer experiment from *E. coli* ErpA to purified IspG orthologs was carried out in an anaerobic chamber ($O_2$ < 2 ppm). For each transfer, one equivalent of apo-IspG (52 nmoles) from *E. coli*, *B. subtilis*, *S. cattleya*, or *Synechocystis sp. CACIAM05* was incubated in transfer buffer (50 mM Tris pH 7.5, 250 mM NaCl, 1 mM DTT) for 5 min before addition of 2.2 equivalents of [Fe-S]-ErpA from *E. coli* (115 nmoles). After 1 hr incubation, the mixture was loaded onto a 5 mL Ni-NTA column. Fractions containing [Fe-S]-IspG were collected in the FT whereas ErpA was eluted using elution buffer (50 mM Tris pH 7.5, 250 mM NaCl, 300 mM imidazole). Fractions were analyzed by SDS-PAGE. UV-visible spectra of ErpA and IspG were recorded before and after incubation and Fe-S transfer.

## TsrM overexpression and purification for in vitro studies

The coding sequence of TsrM from *S. laurentii* preceded by an N-terminal TEV protease cleavable His-tag, was ordered codon optimized and subcloned into pET28a(+) vectors (Genscript). TsrM was coexpressed together with the Btu operon in BL21(DE3) cultured in LB medium supplemented with $FeCl_3$ (50 µM), L-cysteine (150 µM), and hydroxy-cobalamin (2 µM). Btu operon expression was induced at $OD_{600}$ = 0.3 using aTc (100 ng/mL) and TsrM expression was induced using IPTG (1 mM) at an $OD_{600}$ = 0.7 for 18 hr at 18°C. Cells were harvested (6000 rpm, 20 min, 4°C), washed with NaCl 0.9%, and stored at –80°C. TsrM was purified purification under anaerobic conditions ($O_2$ <2 ppm) in a glove box (Jacomex). Cells expressing TsrM were resuspended in buffer containing 50 mM Tris pH 7.5, 250 mM NaCl, 10 mM imidazole, 10% glycerol, 0.1% Tween 20, and sonicated for 7 min (10/30 s ON/OFF, power 50%). After ultracentrifugation (40,000 $g$, 20 min, 4°C), the soluble fraction was loaded onto a 5 mL Ni-NTA column equilibrated with buffer containing 50 mM Tris pH 7.5, 250 mM NaCl. After an extensive washing with buffer containing 50 mM Tris pH 7.5, 250 mM NaCl, 20 mM imidazole buffer,

TsrM was eluted with buffer containing 50 mM Tris pH 7.5, 250 mM NaCl, 300 mM imidazole. Imidazole was removed using a HiPrep 26/10 desalting column equilibrated with buffer containing 50 mM Tris pH 7.5, 250 mM NaCl, 1 mM DTT. TsrM was then concentrated using an amicon cell until 37 mg/mL. As-isolated TsrM was then buffer exchanged into 25 mM Hepes pH 7.5, 300 mM KCl, 5% glycerol using micro Bio-spin six desalting column and used for electrochemistry (cyclic voltammetry) under anaerobic conditions or aliquoted for storage in liquid $N_2$.

## Expression and purification of *S. cattleya* ferredoxins

Coding sequences of the three ferredoxins from *S. cattleya* (AEW96347.1 (Fdx1), AEW92689.1 (Fdx2), AEW97532.1 (Fdx3)), all preceded by an N-terminal TEV protease cleavable His-tag, were ordered codon optimized for expression in *E. coli* and subcloned into pET28a(+) vectors (Genscript). Each ferredoxin was expressed in BL21(DE3) cells cultured in LB medium. Protein expression was induced using IPTG (1 mM) for 3 hr at 37°C. Cells were harvested (6000 rpm, 20 min, 4°C) and washed with NaCl 0.9% before storage at –80 °C. All ferredoxins were purified under anaerobic conditions (0$_2$ <2 ppm) in a glove box. Cells expressing ferredoxins were resuspended in buffer containing 50 mM Tris pH 7.5, 250 mM NaCl, 10 mM imidazole, 10% glycerol, 0.1% Tween 20, and sonicated for 7 min (10/30 s ON/OFF, power 50%). After ultracentrifugation (40,000 *g*, 20 min, 4°C), the soluble fractions were loaded onto three separate 5 mL Ni-NTA columns equilibrated with 50 mM Tris pH 7.5, 250 mM NaCl. Columns were washed using buffer containing 50 mM Tris pH 7.5, 250 mM NaCl, 20 mM imidazole, and proteins were eluted with buffer containing 50 mM Tris pH 7.5, 250 mM NaCl, 300 mM imidazole. Imidazole was removed using a HiPrep 26/10 desalting column, equilibrated with a buffer containing 50 mM Tris pH 7.5, 250 mM NaCl, 1 mM DTT. Ferredoxins cleared of imidazole were incubated overnight in presence of TEV (molar ratio: 1/100). The mixtures were reloaded onto five different 5 mL Ni-NTA columns and FT, containing ferredoxins, were collected.

## Fe-S reconstitution of *S. cattleya* Fdx1

Fdx2 and Fdx3 from *S. cattleya* were purified anaerobically with their Fe-S clusters in contrast to Fdx1 which was obtained as an apo-form. Fdx1 was therefore reconstituted within an anaerobic chamber. Fdx1 was incubated in buffer containing 50 mM Tris pH 7.5, 250 mM NaCl, 1 mM DTT with 10-fold equivalents of $Fe^{2+}$ (Mohr salt) and 10-fold equivalents of $S^{2-}$ (Na$_2$S). The reaction occurred for 3 hr at RT and after centrifugation (15 min at 16,000 *g*) the mixture was loaded onto a Superdex-75 10/300 column. Fractions containing reconstituted Fdx1 were pooled and concentrated using concentrators for microfuge. Subsequent analysis of Fe content and concentration determination were performed as described below.

## Biochemical analyses of TsrM

Concentration of proteins (TsrM and ferredoxins) was determined using Rose Bengal with BSA as standard (*Elliott and Brewer, 1978*) and the Fe content was determined using the Fish method (*Fish, 1988*). The cobalamin content of as-isolated TsrM was determined by UV-visible absorption spectroscopy through its conversion to dicyanocobalamin ($\varepsilon_{367}$ = 30,800 $M^{-1}cm^{-1}$) using a treatment with 0.1 M potassium cyanide following a procedure previously reported (*Blaszczyk et al., 2016*).

## Protein-film electrochemistry

Protein-film electrochemistry experiments were performed anaerobically inside an anaerobic chamber (O$_2$ <2 ppm) using freshly purified TsrM, [Fe-S]-containing ferredoxins and a potentiostat (biologic). A three-electrode configuration was used in a small volume analytical cell (biologic) with a platinum wire and an Ag/AgCl electrode as counter and reference electrode, respectively. When analyzing TsrM, a pyrolytic graphite edge (PGE) electrode was used to collect electrochemical measurements. The electrode was polished with sand paper 1200 followed by 1 µm alumina, and then baseline measurements were collected by placing the PGE electrode into the buffer cell solution (10 mM MES, 10 mM CHES, 10 mM TAPS, 10 mM HEPES pH 8, 200 mM NaCl). Then, 3 µL of 350 µM TsrM were applied to the polished PGE electrode for 5 min before being rinsed with 200 µL of the buffer cell solution. Next, the electrode was immediately placed back in the buffer cell solution for measurements. For ferredoxins, a glassy carbon electrode polished using 1 µm alumina was used within a setup combining a small volume analytical cell and the sample holder of the SVC-2 kit (Biologic). Ferredoxin solutions were at

50 µM (in 10 mM MES, 10 mM CHES, 10 mM TAPS, 10 mM HEPES pH 8, 200 mM NaCl buffer). Cyclic voltammograms were collected at room temperature with a scan rate of 100 mV/s. Redox potentials of TsrM and ferredoxins were determined through square wave voltammetry (SWV) under the same conditions as previously described for cyclic voltammetry. The SWV input signal consisted of a staircase ramp from −1 to –0.3 V vs Ag/AgCl, with 2 mV increments, 50 mV stair amplitude, and 5 Hz frequency. Electrochemical signals were analyzed by correction of the non-Faradaic component from the raw data using the QSoas package (*Fourmond et al., 2009*).

## TsrM-ferredoxin affinity measurements

Affinity interactions between TsrM and *S. cattleya* ferredoxins were measured by BLI inside an anaerobic chamber ($O_2$ <2 ppm) equipped with a BLItz system (FortéBio). All proteins were buffer exchanged into 25 mM Hepes pH 7.5, 300 mM KCl, 5% glycerol using micro Bio-spin6 desalting columns. Ferredoxins were biotinylated through incubation with one equivalent of NHS-$PEG_4$-Biotin. After 30 min, excess NHS-$PEG_4$-Biotin was removed using a micro Bio-spin six desalting column. Biotinylated ferredoxins (10 µg/mL) were bound for 60 s to streptavidin biosensors equilibrated in BLI buffer (25 mM Hepes pH 7.5, 300 mM KCl, 5% glycerol) containing Tween 20 and BSA to inhibit nonspecific binding, following manufacturer recommendations. After having been plunged for 20 s for equilibration in BLI buffer, the loaded biosensor was plunged into TsrM solution for 80 s, followed by 60 s into BLI buffer for association constant and dissociation constant measurements, respectively. This procedure was repeated for various concentrations of TsrM. Affinity constants were extracted from fitted raw data using the BLItz software (PALL/FortéBio).

## Acknowledgements

We thank members of the Bokinsky Lab and the Barras Unit for discussions throughout the work, and Mohamed Atta and Julien Pérard for their assistance with biochemistry experiments. We also thank Roland Lill for valuable suggestions on the manuscript. This project (IRONPLUGNPLAY) has received funding from the European Union's Horizon 2020 research and innovation programme under grant 722361 and was supported by grants from Agence Nationale Recherche (ANR): ANR-10-LABX-62-IBEID, the LabEx ARCANE (ANR-11-LABX-0003-01), and the CBH-EUR-GS (ANR-17-EURE-0003). Synthetic DNA prepared by the Joint Genome Institute DNA Synthesis Program was provided to GB (awarded proposal 503636). The work conducted by the U.S. Department of Energy Joint Genome Institute, a DOE Office of Science User Facility, is supported under Contract No. DE-AC02-05CH11231. SL was supported by the China Scholarship Council.

## Additional information

### Funding

| Funder | Grant reference number | Author |
|---|---|---|
| Horizon 2020 | ERACoBioTech 722361 (project IRONPLUGNPLAY) | Sandrine Ollagnier de Choudens Frédéric Barras Gregory Bokinsky |
| Agence Nationale de la Recherche | ANR-10-LABX-62-IBEID | Frédéric Barras Simonetta Gribaldo |
| Agence Nationale de la Recherche | ANR-11-LABX-0003–01 | Sandrine Ollagnier de Choudens |
| Agence Nationale de la Recherche | ANR-17-EURE-0003 | Sandrine Ollagnier de Choudens |
| China Scholarship Council | | Siyi Liu |

The funders had no role in study design, data collection and interpretation, or the decision to submit the work for publication.

## Author contributions
Francesca D'Angelo, Data curation, Formal analysis, Investigation, Writing – original draft; Elena Fernández-Fueyo, Conceptualization, Formal analysis, Investigation, Methodology; Pierre Simon Garcia, Conceptualization, Data curation, Formal analysis, Investigation, Methodology, Software, Visualization, Writing – original draft; Helena Shomar, Conceptualization, Formal analysis, Investigation, Methodology, Visualization, Writing – original draft, Writing – review and editing; Martin Pelosse, Formal analysis, Investigation, Methodology, Writing – original draft; Rita Rebelo Manuel, Ferhat Büke, Siyi Liu, Investigation; Niels van den Broek, Nicolas Duraffourg, Carol de Ram, Investigation, Methodology; Martin Pabst, Investigation, Methodology, Resources, Supervision; Emmanuelle Bouveret, Investigation, Methodology, Resources; Simonetta Gribaldo, Conceptualization, Investigation, Methodology, Resources, Software, Supervision, Writing – review and editing; Béatrice Py, Funding acquisition, Investigation, Methodology, Resources, Supervision, Writing – review and editing; Sandrine Ollagnier de Choudens, Conceptualization, Data curation, Funding acquisition, Investigation, Methodology, Resources, Supervision, Writing – original draft, Writing – review and editing; Frédéric Barras, Conceptualization, Data curation, Funding acquisition, Investigation, Methodology, Project administration, Resources, Supervision, Visualization, Writing – original draft, Writing – review and editing; Gregory Bokinsky, Conceptualization, Funding acquisition, Investigation, Methodology, Project administration, Resources, Supervision, Visualization, Writing – original draft, Writing – review and editing

## Author ORCIDs
Francesca D'Angelo (iD) http://orcid.org/0000-0001-5880-1087
Helena Shomar (iD) http://orcid.org/0000-0002-1090-2871
Rita Rebelo Manuel (iD) http://orcid.org/0000-0001-8068-2053
Simonetta Gribaldo (iD) http://orcid.org/0000-0002-7662-021X
Sandrine Ollagnier de Choudens (iD) http://orcid.org/0000-0002-0080-6659
Frédéric Barras (iD) http://orcid.org/0000-0003-3458-2574
Gregory Bokinsky (iD) http://orcid.org/0000-0002-7256-4492

## Decision letter and Author response
Decision letter https://doi.org/10.7554/eLife.70936.sa1
Author response https://doi.org/10.7554/eLife.70936.sa2

---

# Additional files

## Supplementary files
• Supplementary file 1. Accession numbers and nucleotide sequences of heterologous NadA, IspG, and IspD orthologs used in the preliminary screen. Complementation results indicated with minus (-): negative complementation; plus (+): positive complementation; (+/-): growth detected but substantially slower than with *E. coli* ortholog; N.A.: not analyzed.

• Supplementary file 2. List of genomes used for the phylogenetic analysis.

• Supplementary file 3. List of the NadA and IspG orthologs retrieved from the database depicted in *Supplementary file 2*. Sequences that have been selected for the complementation test are indicated by a (+) in the adjacent column.

• Supplementary file 4. Accession numbers and nucleotide sequences used to create a phylogenetically-representative selection of NadA and IspG orthologs.

• Supplementary file 5. List of the Fe-S biogenesis proteins and electron carriers used to test re-activation of IspG orthologs using complementation, split into organisms from the phylogenetically-based set (top) and the preliminary survey set (bottom). Re-activation of IspG orthologs was tested first by co-expressing all genes as a multi-cistronic operon: (+) indicating positive complementation; (-) indicating no complementation observed against any IspG ortholog tested. For cases in which individual electron carriers and Fe-S biogenesis proteins were sub-cloned for complementation testing, proteins activating at least one heterologous IspG ortholog (positive complementation) are indicated by (+), while proteins that did not activate IspG (no complementation) are indicated by (-).

• Supplementary file 6. IspG reactivation using plasmids listed in *Supplementary file 5*, including cross-activation of IspG orthologs by non-native electron carrier proteins. IspG orthologs belonging to the phylogenetically-based set and orthologs from the preliminary survey that were not included

in the phylogenetic set are depicted separately. Minus (-): negative complementation; plus (+): positive complementation; (+/-): growth detected but substantially slower than with *E. coli* ortholog; N.T.: not tested. The rightmost column, provided to summarize reactivation experiments, indicates IspG orthologs that were successfully activated by at least one plasmid.

- Supplementary file 7. Results of SDS-PAGE and shotgun proteomics of selected orthologs. Orthologs detected by visible bands on the gel are indicated by a (+).
- Supplementary file 8. List of the bacterial strains, plasmids, and oligonucleotides used in this study.
- Transparent reporting form

### Data availability

All data generated or analysed during this study are included in the manuscript and supporting files. Source data files have been provided for Figures 2 and 3. Data for phylogenetic analysis (containing protein database of the 248 prokaryotes, NadA/ IspG alignments and phylogenies and python script used to select sequences by clustering approach) is available on figshare.

The following dataset was generated:

| Author(s) | Year | Dataset title | Dataset URL | Database and Identifier |
|---|---|---|---|---|
| Garcia P | 2021 | Dangelo_et_al_Supplementary_data.zip | https://doi.org/10.6084/m9.figshare.13664927.v1 | figshare, 10.6084/m9.figshare.13664927.v1 |

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
