## [Decision Letter]

**Decision letter after peer review:**

Thank you for submitting your article "Resolving the barriers that impair iron-sulphur cluster enzyme activity in non-native prokaryotic hosts" for consideration by *eLife*. Your article has been reviewed by 3 peer reviewers, including Dennis Dean as Reviewing Editor and Reviewer #1, and the evaluation has been overseen by Gisela Storz as the Senior Editor. The following individual involved in review of your submission has agreed to reveal their identity: Wayne Outten (Reviewer #2).

The reviewers have discussed their reviews with one another, and the Reviewing Editor has drafted the following comments to help you prepare a revised submission: Your manuscript has been reviewed by three individuals that have considerable experience in the general area of Fe-S protein maturation. Although there is general agreement that there are interesting aspects of the study there are several concerns that preclude publication of the work in its present form. Most of these are described in detail in the initial review that is included elsewhere in this communication. In the discussion among the reviewers, they were asked to make specific suggestions about aspects that could be addressed without a demand for additional experiments such that the work might be suitable for publication in *eLife*. Please note that the rather specific suggestions provided here do not represent "conditions" for publication but rather points that should be considered if you and coworkers should decide to submit a revised manuscript. Also please be aware that any revised manuscript will be evaluated by all the participating reviewers.

The title and abstract would benefit from refinement. In the opinion of the reviewers the work does not, in fact, "resolve the barriers that impair Fe-S cluster enzyme activity in non-native prokaryotic hosts". If such barriers were resolved by the study this would suggest that future studies are not required for the field, which is probably not correct. As noted in the reviews there are barriers that have not been considered. A more descriptive title that could be considered and seems more appropriate is something like: "Using cellular selections for different iron-sulfur proteins to evaluate activities of divergent homologs in a non-native prokaryotic host". Such a title that emphasizes the cellular assay aspect would then be more inclusive of the figure 3 data which is about electron carriers rather than enzymes, which seems disconnected to the enzyme narrative highlighted in the title and abstract. It would also be useful if you would consider adjusting the last sentence in the abstract to convey more precisely what can be gleaned from the study.

Some modest further computational analysis of the data set that is more deeply connected to heterologous expression rather than maturation is an aspect the reviewers believe is necessary for a more complete story. You might want to consider an additional figure that involves such an analysis. This would not require additional experimentation but could involve, for example, plotting the calculated translational initiation strength using the Salis model, making some bar graphs that consider source organism thermostability, oxygen conditions, types of biogenesis systems, and relating this to the assay conditions (temperature, oxygen, host Fe-S machinery). Such an analysis would go a long way towards addressing the major concern of the reviewers. Namely, whether other properties correlate with the accumulation of active Fe-S proteins when heterologously expressed other than those directly considered in the manuscript in its present form. Regardless of trends, additional deeper reflection on the data set could be informative for readers to think about with respect to selection pressure on the divergent proteins as they relate assay conditions.

Following this theme, the manuscript might benefit from a bit more set up and discussion on what can be challenging mechanistically for expression and maturation of iron-sulfur proteins in the introduction. The current focus is restricted to challenges involving Fe-S maturation, which seems narrow and somewhat misleading (other potential limitations should be considered such as promoter strength, translation initiation, folding, maturation, etc.). The manuscript might also benefit from reflecting in the discussion on what is needed to further understand the mechanisms responsible for all these trends and what studies are needed in the future to develop design rules for the two proteins studied and other Fe-S family members. This could reflect on the power of the cellular assays used in tandem with expressing different biogenesis systems, adjusting expression of all components and so forth. Please be assured that the reviewers are not trying to guide your work but rather are providing recommendations that we believe would provide a presentation of higher impact such that it is suitable for publication in *eLife*.

We thank you for submitting your manuscript for consideration by *eLife* and hope you will find the reviews and the comments provided useful.

*Reviewer #1 (Recommendations for the authors):*

This reviewer found the work to be thorough and convincing. The rather concise and clear presentation contributes to the value of the manuscript. The only technical question that I would ask is that it is not clear to me whether or not a fully assembled Fe-S protein containing its cognate Fe-S cluster can be heterologously produced but is inactive because the compatible intact electron transfer component is missing, or if the cognate Fe-S cluster is not inserted during heterologous expression because the electron transfer component is missing. I believe the authors touch on this aspect in the manuscript, but it was not entirely clear to me which way this works? Perhaps this issue could be clarified or, perhaps, emphasized?

*Reviewer #2 (Recommendations for the authors):*

The conclusions of the paper are supported by the data presented. This work also provides one of the few systematic analyses of factors that influence heterologous Fe-S protein expression. However, there were several experiments that could have improved the manuscript and helped tease more information out of the studies performed.

1. Analysis of the IspG heterologous expression seems incomplete, especially since (as the authors point out) there are multiple explanations for lack of growth in a genetic complementation assay. It would be helpful for the reader if the authors could provide a Venn diagram or other graphical representation of the IspG family members tested. Such a figure should show which IspG proteins are (a) insoluble or undetectable (b) detectable but not able to reactivated at all, (c) detectable and active in *E. coli*, (d) detectable and active in *E. coli* but only with co-expression of partner electron transfer protein, etc. For example, it was unclear if some of the IspG proteins that were undetectable when expressed alone could be detected when their electron transfer protein was co-expressed with them. It is possible some IspG enzymes are designed to form complexes with their electron donors and may be inherently unstable without them. It was hard to tease that kind of information out of the data as shown.

2. It would have strengthened the paper to have an experiment that directly tests if IspG is receiving the appropriate Fe-S cluster (especially in those cases where the protein was detectable but inactive). Whole-cell EPR or whole-cell Mossbauer seem like appropriate techniques to address this question. Immunoprecipation of IspG after cell labeling with radioactive iron may also be sufficient. The in vitro Fe-S cluster transfer experiment with ErpA on its own is not enough to demonstrate that there is not a problem with Fe-S maturation of IspG in *E. coli*.

3. The last section of the paper on TsrM is interesting and indirectly relevant for the studies on activation of IspG. It would have been perhaps more informative for the authors to pursue the question of redox coupling directly with IspG and partner electron transfer proteins. In addition, it is important to answer the question of whether the partner protein was assisting IspG simply by increasing IspG stability and/or Fe-S cluster loading.

4. For NadA maturation, did the authors attempt more sophisticated analysis to determine if NadA enzymes and their compatible Fe-S biosynthesis pathways may have sequence motifs, different components, or other unique features that would explain why they work better with each other than more distantly related pathways/proteins? That is really a key question. Why do some pathways work better than others when maturing a given Fe-S enzyme? What features of each pathway have specifically diverged during evolution that created barriers between enzymes/maturation pathways?

*Reviewer #3 (Recommendations for the authors):*

As one considers the different mechanisms that could give rise to the observed trends, this study would be strengthened by further analysis.

Some simple things to consider include calculating the translation initiation rate variation using the Salis RBS calculator to determine how this relates to cellular complementation. Additionally, providing insight on how the native host growth temperature (or oxygen conditions) relates to function would beneficial as well. These types of computational analysis would help understand if expression and thermostability might underlie some of the trends observed.

The effect of iron-sulfur cluster assembly systems on rescuing and divergence from the host would benefit from a side-by-side comparison with host systems expressed the same way. Many of these systems have feedback regulation in their native hosts, and it isn't clear if the regulation of synthesizing these systems underlies the complementation observed or the evolutionary distance trends.

In cases where complementation is not observed, it would be interesting to try at lower temperatures (especially if host grows at lower temperature) and under anaerobic conditions where iron-sulfur cluster biogenesis is less burdensome. Additionally, if RBS strength varies, then one can express all of the homologous iron-sulfur proteins as fusions to a N-terminal peptide or GFP to maintain the context of the RBS and ensure more consistent expression conditions when comparing function.

---

## [Author Response]

The title and abstract would benefit from refinement. In the opinion of the reviewers the work does not, in fact, "resolve the barriers that impair Fe-S cluster enzyme activity in non-native prokaryotic hosts". If such barriers were resolved by the study this would suggest that future studies are not required for the field, which is probably not correct. As noted in the reviews there are barriers that have not been considered. A more descriptive title that could be considered and seems more appropriate is something like: "Using cellular selections for different iron-sulfur proteins to evaluate activities of divergent homologs in a non-native prokaryotic host". Such a title that emphasizes the cellular assay aspect would then be more inclusive of the figure 3 data which is about electron carriers rather than enzymes, which seems disconnected to the enzyme narrative highlighted in the title and abstract. It would also be useful if you would consider adjusting the last sentence in the abstract to convey more precisely what can be gleaned from the study.

We did not intend to imply that the barriers were “resolved” in the sense that no further work was needed. As suggested, we changed the title to better describe our work:

“Cellular assays identify barriers impeding iron-sulphur enzyme activity in a non-native prokaryotic host.”

We have also modified the final sentence of the abstract:

“Our results clarify how oxygen sensitivity and incompatibilities with foreign Fe-S and electron transfer networks each impede heterologous activity. In particular, identifying compatible electron transfer proteins and heterologous Fe-S biogenesis pathways may prove essential for engineered Fe-S enzyme-dependent pathways.”

Some modest further computational analysis of the data set that is more deeply connected to heterologous expression rather than maturation is an aspect the reviewers believe is necessary for a more complete story. You might want to consider an additional figure that involves such an analysis. This would not require additional experimentation but could involve, for example, plotting the calculated translational initiation strength using the Salis model, making some bar graphs that consider source organism thermostability, oxygen conditions, types of biogenesis systems, and relating this to the assay conditions (temperature, oxygen, host Fe-S machinery). Such an analysis would go a long way towards addressing the major concern of the reviewers. Namely, whether other properties correlate with the accumulation of active Fe-S proteins when heterologously expressed other than those directly considered in the manuscript in its present form. Regardless of trends, additional deeper reflection on the data set could be informative for readers to think about with respect to selection pressure on the divergent proteins as they relate assay conditions.

As requested, we now include results from additional experiments and analyses to determine whether other properties correlate with Fe-S enzyme activity.

1. We have performed further complementation experiments in anaerobic conditions to explore oxygen sensitivity as a barrier to heterologous activity (lines 158-169).

2. We applied the RBS Calculator to predict translation initiation rates for heterologous orthologs. These values are now included in Tables 1 and 2. The distributions are depicted and compared in Figure 2 —figure supplement 3 and described in lines 251-257. This analysis did not reveal any significant differences between RBS strength distributions for groups of complementing and non-complementing orthologs.

3. We now include the properties of the source organisms of each tested ortholog (oxygen tolerance, growth temperature, Fe-S biogenesis operon) in Tables 1 and 2. The properties are linked to complementation results in new bar graphs shown in Figure 2E.

These experiments and analyses revealed striking trends in our data, which are now discussed:

1. We found a correlation between source organism aerotolerance and complementation in aerobic conditions for NadA orthologs. This further revealed the existence of two clades with in the NadA phylogenetic tree: an aerotolerant clade, and a clade whose members largely required anaerobic conditions for activity. This is described in lines 185-201, and depicted in Figure 2.

2. We found no clear correlation between complementation ability and either thermotolerance or Fe-S biogenesis operons of the source organisms in the initial complementation experiments using the phylogenetically-based sets of orthologs.

3. However, this additional analysis revealed that the NadA orthologs recovered by heterologous SUF expression were entirely obtained from organisms with exclusively SUF-type pathways. This is described in lines 182-184 and depicted in Figure 2E.

4. The remarkable illustration of oxygen sensitivity as a selection pressure driving diversification within the NadA tree is now elaborated further in the Discussion section (lines 350-356).

Following this theme, the manuscript might benefit from a bit more set up and discussion on what can be challenging mechanistically for expression and maturation of iron-sulfur proteins in the introduction. The current focus is restricted to challenges involving Fe-S maturation, which seems narrow and somewhat misleading (other potential limitations should be considered such as promoter strength, translation initiation, folding, maturation, etc.).

Challenges related to heterologous protein expression are certainly relevant to our work. However, we focus upon challenges particular to Fe-S enzymes, rather than the problems related to protein expression. We therefore sought to exclude effects arising from low promoter strength or inefficient translation as much as possible by using codon-optimized genes and expression from multi-copy plasmids, thus ensuring that all Fe-S enzymes are expressed. We furthermore confirmed expression of a selection of heterologous orthologs using SDS-PAGE and mass spectrometry. The additional analysis by the RBS calculator, which found no evidence that inactive orthologs suffer from inefficient translation, supports our approach. This allows us to identify barriers to activity that are particular to Fe-S enzymes, such as incompatibility with Fe-S biogenesis pathways, oxygen sensitivity, and a lack of suitable electron transfer proteins.

We have modified the text to more clearly acknowledge that problems with heterologous expression may also impede Fe-S enzyme activity:

1. We broadened the introduction to encompass the challenges of heterologous protein expression before emphasizing that we are narrowly focusing upon barriers to activity that are specific to Fe-S enzymes (lines 76-83).

2. We emphasize our use of codon optimization and multicopy plasmids to prevent poor expression from influencing our complementation assays in the Results section (lines 106-109)

3. As explained above, we describe using the RBS Calculator and untargeted proteomics assays to confirm expression of the orthologs found to be inactive (Figure 2 —figure supplement 3, lines 251-257).

The manuscript might also benefit from reflecting in the discussion on what is needed to further understand the mechanisms responsible for all these trends and what studies are needed in the future to develop design rules for the two proteins studied and other Fe-S family members. This could reflect on the power of the cellular assays used in tandem with expressing different biogenesis systems, adjusting expression of all components and so forth. Please be assured that the reviewers are not trying to guide your work but rather are providing recommendations that we believe would provide a presentation of higher impact such that it is suitable for publication in eLife.

We have expanded the discussion to reflect upon further experiments needed for identifying mechanisms. We further elaborate on the insight provided for pathway engineers, and on the additional studies required for formulating reliable “design rules” for Fe-S enzymes:

1. Based on our new findings revealing aerotolerance as a key factor in heterologous NadA activity, we propose experiments to explore the structural basis of aerotolerance in NadA (lines 354-356).

2. We also propose that identifying clades within Fe-S enzyme families (as we did for NadA) may reveal the existence of aerotolerant clades more compatible with aerobic growth (lines 405-410).

3. We also emphasize that some Fe-S enzymes may require heterologous Fe-S biogenesis pathways. Our results suggest using heterologous SUF pathways for enzymes obtained from organisms with SUF pathways (lines 410-414).

4. We explicitly link the need for additional studies that test our suggested guidelines to the eventual formulation of design rules for Fe-S enzymes (lines 420-422).

We thank you for submitting your manuscript for consideration by eLife and hope you will find the reviews and the comments provided useful.Reviewer #1 (Recommendations for the authors):This reviewer found the work to be thorough and convincing. The rather concise and clear presentation contributes to the value of the manuscript. The only technical question that I would ask is that it is not clear to me whether or not a fully assembled Fe-S protein containing its cognate Fe-S cluster can be heterologously produced but is inactive because the compatible intact electron transfer component is missing, or if the cognate Fe-S cluster is not inserted during heterologous expression because the electron transfer component is missing. I believe the authors touch on this aspect in the manuscript, but it was not entirely clear to me which way this works? Perhaps this issue could be clarified or, perhaps, emphasized?

We thank the reviewer for the positive feedback. To address the reviewer’s question about whether electron transfer proteins assist either Fe-S cluster transfer or enzyme activity, we tested whether an IspG ortholog unable to complement growth contained an Fe-S cluster when expressed by *E. coli*. IspG from S. thermophilum was expressed in aerobic cultures and purified under anaerobiosis. in vitro assays (UV-visible spectra, together with iron and sulfur quantification) revealed that a significant fraction of purified IspG contained an Fe-S cluster. This suggests that this particular IspG ortholog is inactive because its cognate electron transport protein is missing, instead of an inability to acquire an Fe-S cluster. The missing electron transfer protein may be required during the IspG catalytic cycle or for cluster maturation step.

We have modified our manuscript to incorporate these results and to clarify our Discussion:

1. Our new characterization of purified Fe-S enzymes are now described in the new version lines 278-291 and Figure 2 —figure supplement 4. Further experiments along these lines with NadA orthologs are described below in our response to a similar question from Reviewer 2.

2. We now acknowledge in our manuscript that electron transfer proteins may be required for IspG cluster maturation, activity, or reduction after adventitious oxidation (lines 374-376 in Discussion).

3. We have removed our assertion that inactive IspG orthologs possess Fe-S clusters. While our new data supports this assertion for S. thermophilum IspG, we have not tested additional inactive IspG orthologs.

4. We have clarified that our observation that an electron transfer protein is sufficient to reactivate many IspG orthologs indicates that activities of these orthologs are not limited by any step facilitated by SUF or ISC Fe-S biogenesis and maturation pathways (lines 366-369 in Discussion). This is true even if the missing electron transfer proteins are required for Fe-S cluster transfer or further maturation.

Reviewer #2 (Recommendations for the authors):The conclusions of the paper are supported by the data presented. This work also provides one of the few systematic analyses of factors that influence heterologous Fe-S protein expression. However, there were several experiments that could have improved the manuscript and helped tease more information out of the studies performed.1. Analysis of the IspG heterologous expression seems incomplete, especially since (as the authors point out) there are multiple explanations for lack of growth in a genetic complementation assay. It would be helpful for the reader if the authors could provide a Venn diagram or other graphical representation of the IspG family members tested. Such a figure should show which IspG proteins are (a) insoluble or undetectable (b) detectable but not able to reactivated at all, (c) detectable and active in E. coli, (d) detectable and active in *E. coli* but only with co-expression of partner electron transfer protein, etc.

To clarify analysis of the IspG orthologs, we made several modifications to the text and tables. We have also added new graphical representations to Figures:

1. We now include a pie chart in Figure 2E depicting the numbers of IspG orthologs active, recovered using electron carriers, recovered in anaerobic conditions, and not recovered.

2. In Tables 1 and 2, we now indicate NadA and IspG orthologs analyzed and detected by MS.

3. In Table 2, we have consolidated data from electron transfer protein recovery experiments into a single column to more cleanly indicate the IspG orthologs that were recovered. Information on the specific electron carriers activating the IspG orthologs has been moved to Supplementary File 6.

4. We provide a pie chart in Figure 2 —figure supplement 3 that depicts the numbers of inactive IspG orthologs that were analyzed using MS, and which of these were successfully detected. This pie chart better clarifies that our MS detection focused upon the inactive IspG orthologs, as MS experiments were motivated by our need to confirm that inactive orthologs were successfully expressed. For orthologs that were detected and undetected, the characteristics of the native hosts are displayed.

5. We emphasize in the text that our MS analysis was focused primarily upon orthologs unable to complement in any condition tested (lines 260-266).

For example, it was unclear if some of the IspG proteins that were undetectable when expressed alone could be detected when their electron transfer protein was co-expressed with them. It is possible some IspG enzymes are designed to form complexes with their electron donors and may be inherently unstable without them. It was hard to tease that kind of information out of the data as shown.

We did not directly test whether expression of compatible electron transfer proteins stabilizes IspG orthologs. This is because our mass spectroscopy experiments were primarily motivated by our need to confirm that orthologs inactive in our complementation assays were expressed. We therefore focused upon orthologs that did not complement growth in any condition. Therefore, the only orthologs we submitted to mass spec analysis were those that did not show any evidence of expression, i.e. because they did not complement growth either alone in aerobic conditions or when co-expressed with a SUF pathway or an electron transfer protein. As stated above, we now clarify that MS was used primarily to detect inactive orthologs on lines 260-263.

Nevertheless, our prior mass spec results can partly address the reviewer’s question of whether inactive IspG can be detected in the absence of suitable electron donors. Aerobic cultures expressing 16 inactive IspG orthologs were analyzed by mass spec. As 10 of the 16 orthologs were detected by mass spec without co-expression of any compatible electron transfer protein, it can be concluded that these orthologs apparently do not require electron transfer proteins for protein stability. We cannot state whether the 6 orthologs analyzed but not detected by mass spec would be detected had we been able to identify and co-express compatible electron transfer proteins.

However, complementation experiments in anaerobic conditions (performed for the revised manuscript) revealed that one of the 6 IspG orthologs that could not be detected by MS (from Collinsella stercoris) recovered growth when expressed under anaerobic conditions. This implies that some IspG orthologs may indeed be unstable (or insoluble) in aerobic conditions, which would make them less detectable by MS. This observation is highlighted on lines 272-276.

2. It would have strengthened the paper to have an experiment that directly tests if IspG is receiving the appropriate Fe-S cluster (especially in those cases where the protein was detectable but inactive). Whole-cell EPR or whole-cell Mossbauer seem like appropriate techniques to address this question. Immunoprecipation of IspG after cell labeling with radioactive iron may also be sufficient. The in vitro Fe-S cluster transfer experiment with ErpA on its own is not enough to demonstrate that there is not a problem with Fe-S maturation of IspG in *E. coli*.

We agree, and we address a nearly identical question in our response to Reviewer 1. While equipment limitations related to the COVID pandemic precluded a timely whole-cell EPR analysis, we have performed an additional biochemical analysis on a subset of NadA and IspG proteins, which is now presented in the revised manuscript (lines 278-291 and Figure 2 —figure supplement 4).

We tested whether several NadA and IspG proteins that scored negative in the aerobic complementation test contained an Fe-S cluster within aerobic *E. coli* cultures. After anaerobic purification, UV-visible spectroscopy together with iron and sulfur quantification (by both colorimetric method and ICP-MS) indicated that at least a small fraction of each of the 4 enzymes purified contained an Fe-S cluster. This indicates that each of these 4 orthologs receive Fe-S clusters. In the case of D. thermophilum and D. indicum NadA orthologs and T. maritima IspG, each of which complement growth in anaerobic conditions, the lack of complementation during aerobic conditions leads us to speculate that the Fe-S clusters detected may be oxidized or otherwise inactivated. In the case of S. thermophilum IspG, which is inactive even in anaerobic conditions, the detection of an Fe-S cluster further suggests that this particular ortholog is inactive because its cognate electron transfer protein is missing, not because it cannot receive a cluster. These new data are now described in the new version lines 287-291 and in Figure 2 —figure supplement 4.

3. The last section of the paper on TsrM is interesting and indirectly relevant for the studies on activation of IspG. It would have been perhaps more informative for the authors to pursue the question of redox coupling directly with IspG and partner electron transfer proteins.

Examining redox coupling between IspG and electron transfer proteins would indeed be more relevant to IspG complementation. However, this would require additional experiments that may not reveal much that is not already known, as the potentials for several electron transfer proteins that we used to activate heterologous IspG have been measured and published. We have now included these published redox potentials in an expanded Discussion (lines 385-393) in which we compare the potentials to in vitro measurements of IspG activity. Given that the redox potentials of FldA and the electron transfer proteins used to activate heterologous IspG are very similar, it is likely that selectivity for specific IspG is driven by structural recognition between IspG and electron transfer proteins, rather than by redox potential.

In addition, it is important to answer the question of whether the partner protein was assisting IspG simply by increasing IspG stability and/or Fe-S cluster loading.

The reviewer is correct that we cannot distinguish for most IspG orthologs whether the electron transfer proteins recover IspG activity by providing electrons for the IspG catalytic cycle (as we originally stated), or whether recovery is enabled through Fe-S cluster loading, maturation, or stability. We have therefore clarified that electron transfer proteins may contribute to any of these steps in the discussion (lines 374-376).

While we cannot conclusively answer this question for all IspG orthologs tested, our experiment with S. thermophilum IspG, which remained inactive in our complementation assays and yet retained an Fe-S cluster following anaerobic purification from aerobic cultures of *E. coli*, suggests that an activating electron transfer protein is not absolutely required for Fe-S cluster loading (lines 277-281). Furthermore, as we state above, mass spectroscopy detection of 10 of 16 IspG orthologs that are inactive in aerobic conditions indicates that electron transfer proteins are not necessarily needed to stabilize the IspG protein itself. On the other hand, the Collinsella stercoris IspG ortholog was not detected in samples from an aerobic culture, but nevertheless recovered growth in an anaerobic culture. This may indicate that some oxygen-sensitive IspG orthologs may be unstable or insoluble during aerobic growth (lines 272-276).

4. For NadA maturation, did the authors attempt more sophisticated analysis to determine if NadA enzymes and their compatible Fe-S biosynthesis pathways may have sequence motifs, different components, or other unique features that would explain why they work better with each other than more distantly related pathways/proteins? That is really a key question. Why do some pathways work better than others when maturing a given Fe-S enzyme? What features of each pathway have specifically diverged during evolution that created barriers between enzymes/maturation pathways?

We agree, and we suggest a similar study that would investigate the sequence motifs and structures underlying the aerotolerance of NadA orthologs in the Discussion (lines 354-356). Concerning motifs that influence compatibility between Fe-S pathways and NadA orthologs, we are currently performing such an in-depth analysis that combines bioinformatics with structural and functional analysis. The data will be published in a separate publication.

Reviewer #3 (Recommendations for the authors):As one considers the different mechanisms that could give rise to the observed trends, this study would be strengthened by further analysis.Some simple things to consider include calculating the translation initiation rate variation using the Salis RBS calculator to determine how this relates to cellular complementation. Additionally, providing insight on how the native host growth temperature (or oxygen conditions) relates to function would beneficial as well. These types of computational analysis would help understand if expression and thermostability might underlie some of the trends observed.

We have performed additional experiments and analysis requested (as also described in our response to the Editor):

1. In order to explore how variations in translation initiation rate relate to the trends we observed, we used the RBS calculator (Salis et al.,) to compare the predicted translation initiation rates of complementing and non-complementing orthologs. No significant differences between RBS strength distributions of complementing and non-complementing orthologs could be found. These analyses are presented in Tables 1 and 2, depicted in Figure 2 —figure supplement 3, and described on lines 251-257.

2. As described in our response to the Editor, we have also analyzed whether growth temperature differences between *E. coli* and sources of the heterologous orthologs, the presence of oxygen, and presence of ISC or SUF pathways relate to function in *E. coli*. These results are depicted in Figure 2E, and the data provided in Table 2. We did not find any clear relationship between growth temperature and complementation.

3. Additional complementation results in anaerobic conditions revealed the existence of two clades within the NadA phylogeny – one of which is aerotolerant, the other likely oxygen-sensitive. Interestingly, membership of NadA orthologs within either clade better predicts complementation in aerobic conditions that does the oxygen tolerance of the original host organisms. This is more fully described in lines 158-169 and lines 185-201.

4. While we found no strong relationship between presence of ISC or SUF pathways in original hosts and activity within *E. coli*, we did find that re-activation of NadA orthologs by heterologous SUF pathways was only possible if the source organism also possessed a SUF pathway. This is described in lines 182-184 and in the Discussion (lines 412-414).

The effect of iron-sulfur cluster assembly systems on rescuing and divergence from the host would benefit from a side-by-side comparison with host systems expressed the same way. Many of these systems have feedback regulation in their native hosts, and it isn't clear if the regulation of synthesizing these systems underlies the complementation observed or the evolutionary distance trends.

Considering our NadA reactivation experiments with SUF operons, both *E. coli* and *B. subtilis* SUF operons were cloned into identical plasmid backbones (pBbA5a) and expressed using an IPTG-inducible promoter, thus precluding any differences in regulation from affecting our results.

For our initial complementation experiments, we acknowledge that the feedback-limited expression of the *E. coli* ISC pathway encoded in the genome, in contrast with the overexpression of the SUF pathways from multi-copy plasmids, may preclude a side-by-side comparison. This is why we do not compare the compatibility of the *E. coli* ISC pathway against the SUF pathways that we tested.

In cases where complementation is not observed, it would be interesting to try at lower temperatures (especially if host grows at lower temperature) and under anaerobic conditions where iron-sulfur cluster biogenesis is less burdensome. Additionally, if RBS strength varies, then one can express all of the homologous iron-sulfur proteins as fusions to a N-terminal peptide or GFP to maintain the context of the RBS and ensure more consistent expression conditions when comparing function.

As described in our responses to other reviewers, we repeated our experiments in anaerobic conditions. This revealed interesting relationships between oxygen sensitivity and activity, which are detailed in our manuscript.

We repeated our complementation assays at low temperature (28°C). This did not change the outcome (lines 145-146). We do not see a clear relationship between optimal growth temperature of each source organism and complementation results (Figure 2E).

Finally, we acknowledge that N-terminal fusions to our Fe-S enzymes together with a consistent RBS would likely ensure a more consistent expression level between orthologs; however, such fusions might affect enzyme activity and complicate our results. As we did not observe any differences in the RBS strength distributions obtained for active and inactive orthologs, we are confident that our results are not driven by differences in RBS strength.